# EvoMAS: Heuristics in the Loop—Evolving Smarter Agentic Workflows

**Yangbo Wei** [* 1 2]  **Zhen Huang** [* 3 2]  **Ronghao Xu** [3]  **Hong Wang** [3]  **Wei W. Xing** [4]

## Abstract

The rapid development of Large Language Models has driven Multi-Agent Systems (MAS) growth, but constructing efficient MAS requires labor-intensive manual design. Current automation methods generate templated agents, use monolithic optimization, and ignore task complexity gradients. This paper presents Evolutionary MAS (EvoMAS), a biologically-inspired framework whose core is a dynamic and diverse repertoire of seven evolutionary strategies—six biologically-inspired operators (3 exploration, 3 exploitation) together with a custom operator for domain-specific transformations—driven by adaptive strategy selection. These are complemented by role-level evolution that refines agent specialization and collaboration patterns, and a curriculum-guided schedule that evolves workflows from simple to complex tasks with cross-stage stability. Additionally, to resolve the contradiction between the inefficiency of pure evolution and the rigidity of manual design, we introduce the *Cyber Creator*, a meta-controller that conducts heuristics-in-the-loop learning by formulating and reflectively updating evolutionary rules and strategies. Evaluations demonstrate that EvoMAS consistently outperforms existing methods across multiple domains while maintaining cost efficiency, with roles evolving from homogeneous actors to specialized reasoning ensembles.

## 1. Introduction

The rapid advancement of Large Language Models (LLMs), particularly the flourishing ecosystem of Model Context Protocol (MCP) (Hou et al., 2025), has propelled MAS as a powerful collaborative paradigm at the forefront of AI innovation (Li et al., 2024a; Han et al., 2024; Cemri et al., 2025). However, current MAS design methodologies face fundamental challenges: they predominantly rely on static predefined architectures and fixed interaction patterns—a rigid design philosophy that severely constrains their ability to respond to complex and dynamic environments. While such systems may excel in specific scenarios, they exhibit notable *adaptation barriers* when confronting open-ended, dynamic problems. Consequently, the automation and optimization of MAS design has emerged as a critical frontier challenge (Weyns & Oquendo, 2019).

Recent years have witnessed significant progress in agent system automation technologies, albeit with evident bifurcation trends. One category focuses on single-dimensional optimization: DsPy (Khattab et al., 2024a) and EvoPrompting (Chen et al., 2023a) pioneered automated paradigms in prompt engineering; GPTSwarm (Zhuge et al., 2024) and G-Designer (Zhang et al., 2024a) dedicated efforts to optimizing inter-agent communication protocols; while EvoAgent (Yuan et al., 2024) and AutoAgents (Chen et al., 2023b) explored the possibilities of single-agent self-evolution. Despite their respective strengths, these approaches struggle to achieve system-level breakthroughs due to their localized optimization perspectives. Another category, including ADAS (Hu et al., 2024), AgentSquare (Shang et al., 2024), and AFlow (Zhang et al., 2024b), attempts to expand the design search space, constructing optimized workflows on specific datasets through heuristic search, Monte Carlo tree search (MCTS), or evolutionary algorithms, demonstrating capabilities surpassing manually designed systems. Nevertheless, these methods reveal **severe limitations** when facing cross-domain tasks: (1) they typically employ **singular, fixed optimization strategies** inadequate for diverse task requirements; (2) their system structures and agent profiling remain excessively **template-based**, lacking necessary flexibility and innovation potential; (3) they **disregard the crucial impact of task difficulty gradients** on learning efficiency, resulting in insufficient generalization capabilities in complex scenarios.

Addressing these challenges, we introduce **EvoMAS** —a biologically-inspired framework for automated evolution of multi-agent systems. EvoMAS integrates three inter-

---

[*]Equal contribution [1]Shanghai Jiao Tong University, Shanghai, China [2]Eastern Institute of Technology, Ningbo, China [3]University of Science and Technology of China, Hefei, China [4]The University of Sheffield, Sheffield, United Kingdom. Correspondence to: Yangbo Wei <yangforever@sjtu.edu.cn>, Wei W. Xing <wayne.xingle@gmail.com>.

*Proceedings of the 43^{rd} International Conference on Machine Learning*, Seoul, South Korea. PMLR 306, 2026. Copyright 2026 by the author(s).

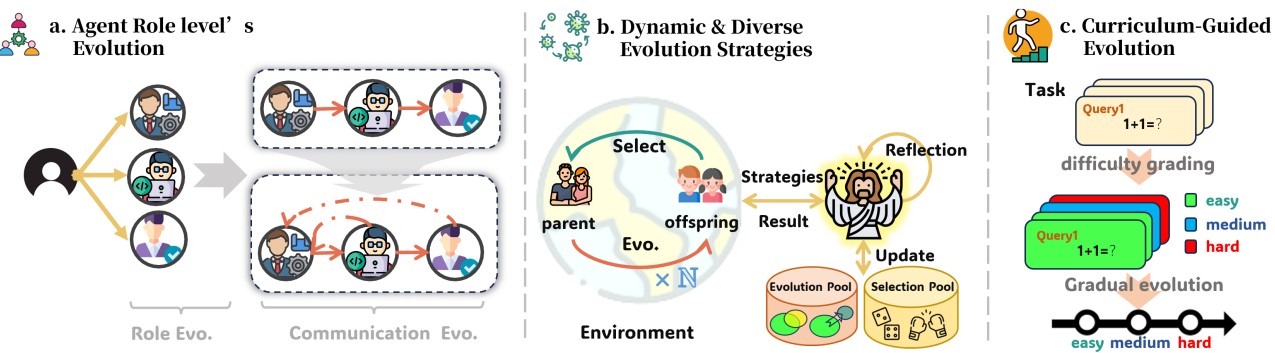

*Figure 1.* Motivation: Three Evolution Dimensions in EvoMAS. EvoMAS explores agentic workflow evolution from three key dimensions: (a) role-level evolution enhances agent specialization and coordination, (b) diverse strategies enable LLM-guided exploration with reflection-based updates, and (c) curriculum-guided evolution promotes gradual adaptation across tasks of increasing difficulty.

connected evolutionary dimensions as shown in Figure 1: ❶ **Role dimension:** the system implements adaptive role evolution mechanisms that dynamically refine agent specializations and interaction patterns throughout the evolutionary process. This transcends traditional rigid agent roles, enabling specialized collaboration tailored to task requirements; ❷ **Strategy dimension:** EvoMAS constructs a dual-track "exploration-exploitation" evolutionary mechanism encompassing six biologically-inspired strategies, enabling the system to achieve an exquisite balance between search space diversity and optimal solution convergence; ❸ **Learning path dimension:** the system implements a curriculum-inspired progressive adaptation process, enabling complex capabilities to build gradually upon simpler tasks, significantly enhancing cross-task generalization abilities.

While natural selection provides powerful optimization principles, purely evolutionary approaches can be inefficient for complex tasks, and conversely, rigid human design often limits its adaptability. To address this fundamental tension, Evo-MAS introduces the *"Cyber Creator"*—a meta-controller that performs heuristics-in-the-loop learning, combining rule-based guidance with adaptive reflection. This mechanism bridges the gap between undirected evolution and artificial intervention through explicit rule-setting and periodic reflective updates. Additionally, EvoMAS employs graph structures to precisely express the topological and functional characteristics of multi-agent workflows, while constructing an evolutionary resource center comprising rule pools and gene pools that provides robust support for knowledge accumulation and transfer. Table 1 illustrates how EvoMAS distinguishes itself from existing automated design methods through comprehensive evolutionary capabilities, being the only framework to simultaneously support dynamic strategy evolution, meta-level adaptation, and curriculum-guided learning. EvoMAS achieves state-of-the-art (SOTA) performance across six benchmarks, while maintaining superior cost-efficiency, outperforming both manual designs and automated baselines.

*Table 1.* **Capability comparison across automated MAS design frameworks.** EvoMAS is the only method that jointly supports multi-agent evolution, dynamic strategy adaptation, meta-level evolution, and curriculum-guided learning.

| Method | MAS Evol. | Dynamic Strategy | Meta-Evol. | Curric. Learn. |
|---|---|---|---|---|
| EvoFlow | ✓ | ✗ | ✗ | ✗ |
| FunSearch | ✗ | ✗ | ✗ | ✗ |
| EvoAgent | ✓ | ✗ | ✗ | ✗ |
| AFlow | ✓ | ✗ | ✗ | ✗ |
| ADAS | ✓ | ✗ | ✗ | ✗ |
| **EvoMAS** | ✓ | ✓ | ✓ | ✓ |

## 2. Related Work

**Agentic Workflow.** With the increasing capabilities of LLMs, the paradigm of Agentic Workflow has emerged as a promising approach to construct structured and multi-stage task-solving processes (Hong et al., 2024a; Zhang et al., 2024c; Wang et al., 2023a). This paradigm typically comprises multiple LLM-invoking nodes with well-defined inputs and outputs, organized in the form of graphs, code, or flowcharts to specify the execution sequence.

Existing research in this area can be broadly categorized into two directions: general-purpose workflows (Madaan et al., 2023; Wang et al., 2023b) and domain-specific pipelines (Zhong et al., 2025; Xu et al., 2024). The former focuses on universal reasoning strategies, while the latter builds tailored structures for specific tasks such as code generation (Ridnik et al., 2024; Hong et al., 2024b), data analysis (Zhou et al., 2023; Ye et al., 2024), and multi-hop question answering (Zhou et al., 2024). However, most approaches rely on predefined templates or operator libraries and lack the expressiveness for hierarchical structural evolution and dynamic adaptation.

**Automated Agentic Optimization.** To alleviate the burden of manually designing complex workflows, research

on automated agentic workflow optimization (Zhuge et al., 2024; Li et al., 2024b; Hu et al., 2024; Zhang et al., 2025) is gaining traction. Some approaches focus on prompt (Khattab et al., 2024b; Fernando et al., 2024; Yang et al., 2024; Yüksekgönül et al., 2024) or parameter tuning (Saad-Falcon et al., 2024), while others aim at optimizing the structural composition of workflows, including inter-module connectivity, execution ordering, and conditional dependencies.

Representative methods include ADAS (Hu et al., 2024), which linearizes workflow code and performs sequential structure search, and GPTSwarm (Zhuge et al., 2024), which models workflows as graphs and uses reinforcement learning for structural optimization. Additionally, AFlow (Zhang et al., 2024b) encodes workflows as code and applies MCTS to explore efficient execution paths, demonstrating superior performance over manual designs. However, these methods still suffer from limited search efficiency, constrained expressiveness, and poor cross-task generalization. In particular, they lack effective mechanisms for heterogeneous module collaboration and feedback-driven structural evolution, limiting their ability to adapt to complex tasks.

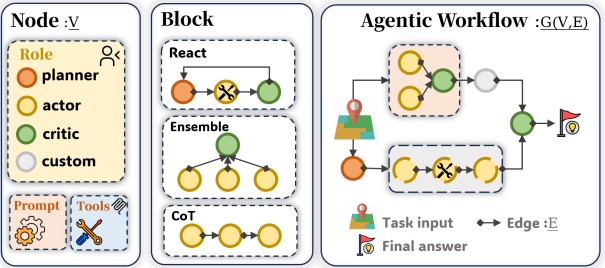

*Figure 2.* Agentic workflow search space represented as graphs with role-specific nodes and modular blocks, enabling dynamic tool use, prompt chaining, and structured multi-agent task solving.

## 3. Preliminary

This section establishes the theoretical foundation for automated multi-agent system design by introducing core design assumptions, formalizing the graph-based structural representation, and defining the constrained optimization problem that guides our evolutionary framework.

### 3.1. Representation: Graph-based Formulation of MAS

To capture the complex control flow, information exchange, and collaborative dynamics inherent in multi-agent systems, we model MAS workflows as sparse, cyclic directed graphs as illustrated in Fig. 2. Formally, a workflow is represented as $G = (V, E)$, where $V = \{v_1, v_2, \ldots, v_n\}$ denotes the vertex set with each node $v_i$ corresponding to an autonomous agent, and $E \subseteq V \times V$ represents the edge set encoding directed dependencies for information flow

and control signal propagation. The graph maintains structural integrity through a unique source node (input) and sink node (output), ensuring well-defined task boundaries and deterministic execution semantics.

### 3.2. Design Assumptions

Our approach to automated multi-agent system evolution is built upon three core assumptions that define the theoretical foundation and constrain the design space:

$\mathcal{H}_1$ **(Heterogeneous Synergy).** *Following Ricardo's comparative advantage theory, specialized agents can achieve higher collective efficiency even when individual capabilities differ.* We assume that heterogeneous agent roles with complementary functional specializations—some focusing on reasoning, others on verification and critique—yield consistently superior performance compared to homogeneous configurations (Bettini et al., 2023; Cao et al., 2019).

$\mathcal{H}_2$ **(Reflective Adaptation).** *Based on Bayesian learning principles ($P(\theta|D) \propto P(D|\theta)P(\theta)$), agents continuously adapt through feedback loops.* We assume that self-monitoring mechanisms enable robust error correction and strategic refinement, allowing systems to maintain stability while adapting to evolving requirements (Bilal et al., 2025).

$\mathcal{H}_3$ **(Structural Parsimony).** *Following Occam's Razor and bias-variance tradeoff principles, simpler architectures often outperform complex ones.* We assume that beyond optimal system size, additional agents introduce coordination overhead with diminishing returns, making structural parsimony essential for efficiency (Narain et al., 2014; Wu et al., 2025).

**Formalization.** These assumptions collectively define our design space $\Omega = \{G | G \text{ satisfies } \mathcal{H}_1 \wedge \mathcal{H}_2 \wedge \mathcal{H}_3\}$, where each candidate workflow $G$ must exhibit role specialization ($\mathcal{H}_1$), incorporate feedback mechanisms ($\mathcal{H}_2$), and maintain structural efficiency ($\mathcal{H}_3$). Our evolutionary framework operates within this constrained space to ensure theoretically grounded and practically viable solutions.

### 3.3. Problem Definition

We formalize the automated design and optimization of multi-agent systems as a constrained single-objective optimization problem over the space of sparse, cyclic directed graphs. Given the design assumptions $\mathcal{H}_1$, $\mathcal{H}_2$, and $\mathcal{H}_3$, the optimization problem is defined as:

$$G^* = \arg\max_{G \in \Omega} F(G, T, R) \quad (1)$$

where $G \in \Omega$ represents a candidate MAS workflow graph constrained by our design assumptions, $T$ denotes the task distribution encompassing problem instances and difficulty metrics, $R = \{r_1, r_2, \ldots, r_k\}$ is the evolutionary rule set

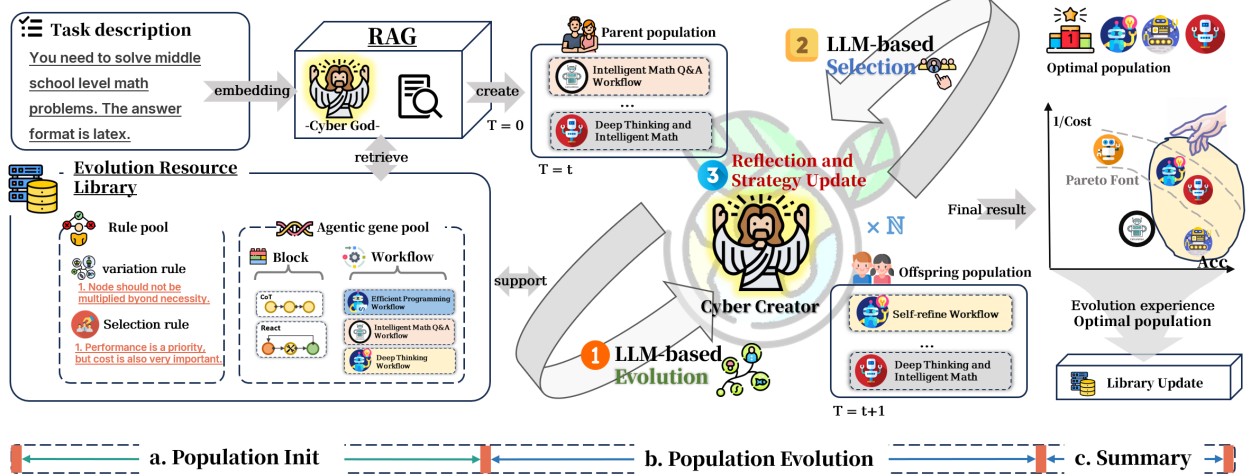

*Figure 3.* **Overall EvoMAS Framework.** The framework follows a three-stage process: (1) initialization via RAG-based retrieval from a knowledge-rich resource library, (2) population evolution through LLM-based variation and selection, periodically guided by the Cyber Creator's reflective rule and strategy updates, and (3) evolution resource library update based on the results.

encoding domain-specific constraints and structural preferences, and $F : \Omega \times \mathcal{T} \times \mathcal{R} \to \mathbb{R}^+$ is the composite fitness function evaluating workflow performance under task distribution $T$ and rule guidance $R$.

This single-objective formulation offers several theoretical advantages over multi-objective approaches: (*i*) it reduces computational complexity by eliminating Pareto frontier approximation; (*ii*) it enables direct application of convergence guarantees from evolutionary optimization theory; and (*iii*) it provides interpretable optimization trajectories through explicit rule-based guidance, facilitating systematic analysis of design trade-offs.

# 4. Methods: EvoMAS

We propose EvoMAS, a biologically-inspired framework for evolving multi-agent workflows. Unlike prior approaches that rely on static architectures or singular optimization strategies, EvoMAS formalizes the agent workflow construction as a rule-guided evolutionary process with stochastic dynamics. This section introduces the core formalism, system components, and theoretical underpinnings. The complete algorithm process is presented in Appendix C.

## 4.1. Evolution as a Markov Process

We model the EvoMAS system as a **non-homogeneous Markov process** over the evolving system state:

$$s_t = (\mathcal{P}_t, R_t, \mathcal{A}_t), \tag{2}$$

where $\mathcal{P}_t$ is the population of agentic workflows, $R_t$ is the rule set (e.g., human- or LLM-injected), and $\mathcal{A}_t$ denotes the current evolution strategies for exploration and exploitation. The evolution dynamics are governed by a probabilistic

transition kernel:

$$P(s_{t+1} \mid s_t) = P\left((\mathcal{P}_{t+1}, R_{t+1}, \mathcal{A}_{t+1}) \mid (\mathcal{P}_t, R_t, \mathcal{A}_t)\right). \tag{3}$$

This formulation encapsulates the full evolution loop—*variation*, *selection*, and *meta-reflection*—under a unified stochastic process.

## 4.2. Evolution Cycle: Variation → Selection → Reflection

EvoMAS operates in discrete generations, where each generation $t$ performs a full evolutionary cycle on the system state $s_t = (\mathcal{P}_t, R_t, \mathcal{A}_t)$. This cycle comprises three interlinked stages: *variation*, *selection*, and *reflection*.

**Variation.** We formalize variation as a set of graph transformation operators $\mathcal{O} : \mathcal{G} \to \mathcal{G}$, designed to navigate the trade-off between exploring novel topologies and exploiting known high-performing structures. EvoMAS implements this through two complementary mechanisms:

**Structural Exploration ($\mathcal{X}$ Strategies).** To escape local optima, exploration operators introduce topological diversity by navigating the discrete graph space $\mathbb{G}$. $\mathcal{X}_1$ (**Diversity Expansion**) mimics *genetic mutation* via stochastic perturbation. It applies a symmetric difference operation, formulated as $G' = (V \Delta V_\delta, E \Delta E_\delta)$, to reversibly inject or prune task-specific nodes, effectively expanding the search envelope. $\mathcal{X}_2$ (**Conceptual Recombination**) mirrors *sexual recombination* by performing a disjoint union of functional subgraphs from parents $G_a$ and $G_b$. This operator merges distinct modules and establishes new connectivity through bridging edges $E_{\text{bridge}}$. $\mathcal{X}_3$ (**Cross-domain Hybridization**) facilitates knowledge transfer via *structural projection*. In-

spired by horizontal gene transfer, it injects motifs from a reference domain $G^{ref}$ into the current graph, modeled as $G' = G \oplus_\alpha \psi(G^{ref})$, where $\psi$ aligns the structural priors.

**Functional Exploitation ($\mathcal{Y}$ Strategies).** Once promising topological manifolds are identified, exploitation operators refine the agents' internal parameter space $\Phi$ without altering the global structure. $\mathcal{Y}_1$ **(Fine Optimization)** applies local search to optimize node parameters (e.g., prompts). It follows a pseudo-gradient trajectory $\Phi' = \Phi - \eta \hat{\nabla} L$, using feedback estimates to approximate differentiable updates. $\mathcal{Y}_2$ **(Best Practice Synthesis)** goes beyond simple selection by computing the *structural centroid* of the elite population $\Omega$. It synthesizes a robust workflow $G' = \bigcup(G_k \cap \mathcal{K}_{\text{core}})$ by consolidating high-frequency substructures shared among top performers. $\mathcal{Y}_3$ **(Role Specialization)** enhances modular efficiency through *function differentiation*. It applies a specialization mapping $\rho' = \mathcal{M}(\rho)$ to transform generic roles into specialized ones, minimizing task ambiguity.

Detailed mathematical formulations and algorithmic procedures for these operators are provided in Appendix B.2.

Formally, each candidate graph $G' \in \mathcal{P}_t^{\text{var}}$ is generated by sampling a parent $G \in \mathcal{P}_t$, a strategy $a \sim \mathcal{A}_t$, and applying the associated transformation operator:

$$G' \sim a(G, R_t), \quad \text{where } a \sim \mathcal{A}_t, G \sim \mathcal{P}_t. \quad (4)$$

where $\mathcal{A}_t = \{\mathcal{X}_1, \mathcal{X}_2, \mathcal{X}_3, \mathcal{Y}_1, \mathcal{Y}_2, \mathcal{Y}_3, \mathcal{C}\}$ encompasses the six biologically-inspired strategies plus a custom strategy $\mathcal{C}$ that enables domain-specific transformations tailored to particular requirements.

**Selection.** After variation, all candidate workflows are evaluated using a base fitness score. Formally, for each workflow graph $G$, we compute its task-level performance via:

$$F(G, T) = \mathbb{E}_{x \sim T}[f(G, x)], \quad (5)$$

where $f(G, x)$ measures the accuracy or success rate of workflow $G$ on input $x$. Rather than relying solely on scalar fitness ranking, we adopt a preference-guided selection mechanism that holistically evaluates candidate workflows based on performance metrics, structural properties, and rule alignment. Specifically, each candidate $G \in \mathcal{P}_t^{\text{var}} \cup \mathcal{P}_t$ is serialized into a compact textual representation that captures three core aspects: 1) its execution performance $F(G, T)$, representing quantitative task success; 2) its internal structure, including the graph topology, node roles (e.g., Planner, Critic), and configuration parameters; 3) its degree of compliance with the current rule set $R_t$, including both hard constraints and soft preferences.

The resulting representations are provided as input to a large language model, which performs *preference-based selection* by implicitly evaluating each candidate according to a latent

utility function informed by human-aligned inductive biases. Rather than relying solely on scalar fitness scores, the LLM considers a richer combination of behavioral performance, structural plausibility, and rule conformance. Based on this holistic assessment, the LLM ranks candidates and selects a subset to form the next generation:

$$\mathcal{P}_{t+1} = \text{LLMSelect}(\mathcal{P}_t \cup \mathcal{P}_t^{\text{var}}, F, R_t), \quad (6)$$

This approach enables EvoMAS to incorporate qualitative notions of agent design (e.g., modularity, interpretability, domain alignment) that are difficult to encode in a scalar objective. We define a best-so-far fitness:

$$B_t := \max_{i \leq t} \max_{G \in \mathcal{P}_i} F(G), \quad (7)$$

which remains non-decreasing due to implicit elitism (high-performing candidates are rarely discarded) and converges under boundedness assumptions.

**Reflection (Cyber Creator).** Every $K$ generations, EvoMAS invokes the *Cyber Creator*, a meta-controller that reformulates workflow generation as a **Bi-level Optimization** problem. While the inner loop performs structural variation on the workflow $G$, the Cyber Creator optimizes the search policy $\pi_t$ itself. We decompose the policy $\pi_t(G) \propto \mathbb{I}(G \in \Omega_{R_t}) \cdot P_{\mathcal{A}_t}(\text{op})$ into a hard constraint manifold $\Omega_{R_t}$ determined by rules and a soft sampling distribution $P_{\mathcal{A}_t}$ determined by strategies. The meta-objective is to maximize the expected improvement (EI) of the evolutionary trajectory:

$$\pi_{t+1} = \arg\max_\pi \mathbb{E}_{G \sim \pi} \left[ \underbrace{J(G)}_{\text{Fitness}} - \lambda \cdot \underbrace{\mathcal{C}(G)}_{\text{Cost}} \right], \quad (8)$$

This meta-optimization is modeled as a Bayesian posterior update $P(\pi|\mathcal{H}_t) \propto P(\mathcal{H}_t|\pi)P(\pi)$. Since the true posterior is intractable, we leverage the LLM as an *approximate inference kernel* to perform the update through two complementary mechanisms:

**1. Discrete Manifold Reshaping (Rule Update via Constraint Synthesis).** The system synthesizes constraints to prune invalid search regions (where prior $P(G) \approx 0$) while generalizing successful patterns into inductive biases. This effectively updates the feasible manifold $\Omega_{R_t}$:

$$R_{t+1} = \mathcal{U}_R(R_t; H_t) = Prune(R_t, H_t) \cup Induce(H_t). \quad (9)$$

This step reduces sample complexity by transforming the optimization problem from an unconstrained search over the entire graph space $\mathbb{G}$ to a constrained search over a high-probability subspace $\Omega_{R_{t+1}}$.

**2. Continuous Strategy Adaptation (Posterior Approximation).** The strategy distribution $\mathcal{A}_t$ undergoes a soft

update using the **Multiplicative Weights Update (MWU)** algorithm. Strategies are re-weighted based on their contribution to the regularized objective defined in Eq. 8:

$$\mathcal{A}_{t+1}(a) = \frac{\mathcal{A}_t(a) \cdot \exp\left(\eta \cdot \hat{r}_t(a)\right)}{\sum_{a'} \mathcal{A}_t(a') \cdot \exp\left(\eta \cdot \hat{r}_t(a')\right)}, \quad (10)$$

where $\hat{r}_t(a)$ estimates the marginal gain in the regularized objective $\Delta(J - \lambda \mathcal{C})$ attributed to strategy $a$. Through this hybrid update process, EvoMAS effectively closes the loop on self-directed evolutionary cognition, ensuring asymptotic convergence to the Pareto-optimal frontier.

### 4.3. Curriculum-Guided Evolution

In biological evolution, complex organisms emerge through gradual adaptation to increasingly challenging environments. Inspired by this principle, EvoMAS incorporates a curriculum-guided evolutionary process, where agentic workflows evolve progressively—from simpler to more complex tasks—thereby improving learning stability, sample efficiency, and generalization.

**Task Difficulty Layering.** We partition the overall task distribution $T$ into $n$ ordered subsets based on increasing cognitive complexity:

$$T = \{T_1, T_2, \ldots, T_n\}, \quad \text{s.t.} \quad d(T_1) < \cdots < d(T_n), \quad (11)$$

where $d(\cdot)$ denotes a task difficulty function that evaluates each subset $T_i$ based on semantic complexity, required reasoning steps, and domain-specific expertise. To estimate $d(T_i)$, we adopt an LLM-as-a-Judge framework (Gu et al., 2024), which provides difficulty ratings by analyzing input-output complexity, abstraction level, and knowledge dependencies. Each subset $T_i$ defines a curriculum stage with internally consistent difficulty, and evolution proceeds sequentially as the system achieves competence at each level.

**Sequential Evolution with Stability Control.** Evolution proceeds sequentially through difficulty stages, with each stage $T_i$ serving as the training environment until competence threshold is reached. To prevent catastrophic forgetting during stage transitions, we enforce a stability constraint: let $G_k$ denote the best-evolved workflow at stage $k$, then the cumulative performance $J_{k+1}(G_{k+1}) = \frac{1}{k+1} \sum_{i=1}^{k+1} f(G_{k+1}, T_i)$ must exceed $J_{k+1}(G_k)$, ensuring that newly evolved workflows maintain competence on previous stages while adapting to increased complexity.

## 5. Experiment

We evaluate EvoMAS along four dimensions: task performance, deployment cost, component effectiveness, and robustness to LLM-dependent modules. Across mathematical

reasoning, code generation, multi-hop question, embodied interaction, and tool-use benchmarks, we compare Evo-MAS against single-agent baselines, manually designed multi-agent systems, and automated workflow optimization methods. Beyond main accuracy results, we further analyze cost-performance trade-offs, ablation effects, evolutionary role specialization, and sensitivity to weaker open-source models, providing a comprehensive assessment of EvoMAS in terms of effectiveness, efficiency, and robustness.

### 5.1. Experimental Setup

**Datasets and Tasks.** We evaluated EvoMAS on 7 public datasets covering three major domains: (1) Mathematical reasoning: GSM8K (Cobbe et al., 2021) and MATH (Hendrycks et al., 2021). (2) Code generation and language reasoning: HumanEval (Chen et al., 2021) and MBPP (Austin et al., 2021) for code generation, and HotpotQA (Yang et al., 2018) for language understanding. (3) Embodied intelligence tasks: ALFWorld (Shridhar et al., 2020) to evaluate agents' multi-step operation and goal execution abilities in virtual environments, and GAIA (Mialon et al., 2023) to assess agents' tool-using capabilities.

**Baselines.** We compared EvoMAS with three categories of agent benchmarks: (1) Single-agent execution methods, including IO (direct LLM invocation) and CoT; (2) Manually designed multi-agent systems, including MultiPersona, LLM-Debate, and AgentVerse; (3) (Partially or fully) autonomous multi-agent systems, including GPTSwarm, AutoAgents, ADAS, AgentSquare, EvoFlow, and AFlow.

More details on setups are provided in Appendix D.

### 5.2. Experimental Results and Analysis

**Main Results.** EvoMAS demonstrates exceptional performance across diverse benchmark tasks, achieving SOTA results on five out of six benchmarks with an average score of 80.06% as shown in Table 2, outperforming the previous best method AFlow by 3.78% on average while maintaining competitive results on ALFWorld (67.28%).

The performance gains are particularly pronounced in mathematical reasoning, where EvoMAS surpasses the strongest baseline AFlow by 1.37% on GSM8K and 4.96% on MATH, and in code generation, with gains of 1.53% on HumanEval and 2.64% on MBPP, reflecting stronger logical reasoning and problem-solving through collaborative agent interactions. On the challenging GAIA embodied-intelligence benchmark (Table 3), EvoMAS leads across all difficulty levels, improving over the second-best method MaAS by 1.90% on average, with the largest gains on the hardest Level-3 tasks, underscoring its capability in complex reasoning, tool use, and real-world coordination requiring sophisticated agent collaboration. Taken together, these consis-

*Table 2.* **Main Results.** Accuracy (%) and standard deviation (±) across six benchmarks. SDs are estimated from 3 independent runs.
**Red** indicates best, **Blue** indicates second best.

| Method | GSM8K | MATH | HumanEval | MBPP | HotpotQA | ALFWorld | Avg. |
|---|---|---|---|---|---|---|---|
| *Single-agent Baselines* | | | | | | | |
| IO (GPT-4o-mini) | $89.46^{\pm 2.15}$ | $47.11^{\pm 1.82}$ | $85.50^{\pm 2.40}$ | $71.83^{\pm 3.10}$ | $67.60^{\pm 2.85}$ | $38.71^{\pm 4.20}$ | 66.70 |
| CoT | $89.31^{\pm 2.08}$ | $47.93^{\pm 1.95}$ | $87.02^{\pm 2.18}$ | $71.83^{\pm 3.05}$ | $68.10^{\pm 2.92}$ | $39.92^{\pm 4.15}$ | 67.35 |
| *Manually Designed MAS* | | | | | | | |
| MultiPersona | $90.37^{\pm 1.95}$ | $48.26^{\pm 2.10}$ | $88.54^{\pm 2.25}$ | $73.02^{\pm 2.85}$ | $69.30^{\pm 2.58}$ | $39.10^{\pm 4.50}$ | 68.10 |
| LLM-Debate | $90.30^{\pm 2.22}$ | $48.76^{\pm 2.18}$ | $87.78^{\pm 2.30}$ | $72.14^{\pm 3.40}$ | $70.10^{\pm 2.75}$ | $44.68^{\pm 4.80}$ | 68.63 |
| AgentVerse | $90.67^{\pm 2.18}$ | $48.10^{\pm 2.25}$ | $89.31^{\pm 2.12}$ | $73.90^{\pm 2.92}$ | $72.50^{\pm 2.60}$ | $45.03^{\pm 4.65}$ | 69.92 |
| *Automated Agentic Systems* | | | | | | | |
| GPTSwarm | $90.14^{\pm 2.40}$ | $47.27^{\pm 3.55}$ | $90.07^{\pm 2.35}$ | $76.83^{\pm 3.45}$ | $68.10^{\pm 3.42}$ | $53.19^{\pm 4.90}$ | 70.27 |
| AutoAgents | $89.53^{\pm 2.35}$ | $46.61^{\pm 3.48}$ | $86.25^{\pm 2.40}$ | $72.14^{\pm 3.50}$ | $66.80^{\pm 3.38}$ | $46.15^{\pm 4.85}$ | 67.58 |
| ADAS | $87.18^{\pm 2.50}$ | $46.61^{\pm 3.60}$ | $83.97^{\pm 2.55}$ | $67.45^{\pm 3.65}$ | $64.60^{\pm 3.50}$ | $47.66^{\pm 4.95}$ | 66.58 |
| AgentSquare | $89.08^{\pm 1.80}$ | $48.26^{\pm 2.45}$ | $90.83^{\pm 1.78}$ | $80.64^{\pm 2.42}$ | $71.70^{\pm 2.35}$ | $66.42^{\pm 3.80}$ | 74.16 |
| AFlow | $93.39^{\pm 1.55}$ | $55.37^{\pm 2.38}$ | $92.36^{\pm 1.60}$ | $83.57^{\pm 2.35}$ | $73.80^{\pm 2.12}$ | $59.16^{\pm 3.60}$ | 76.28 |
| **EvoMAS (Ours)** | $94.76^{\pm 1.15}$ | $60.33^{\pm 1.22}$ | $93.89^{\pm 1.12}$ | $86.21^{\pm 1.25}$ | $77.90^{\pm 1.20}$ | $67.28^{\pm 3.10}$ | **80.06** |

*Table 3.* **GAIA Benchmark Results.** Success rate (%) and standard deviation across different difficulty levels. EvoMAS achieves SOTA performance, particularly on the hardest Level 3 tasks.

| Method | Level 1 | Level 2 | Level 3 | Average |
|---|---|---|---|---|
| GPT-4o-mini | $7.53^{\pm 1.20}$ | $4.40^{\pm 0.85}$ | $0.00^{\pm 0.00}$ | $4.65^{\pm 0.65}$ |
| AutoGPT | $13.21^{\pm 2.50}$ | $0.00^{\pm 0.00}$ | $3.85^{\pm 1.10}$ | $4.85^{\pm 1.25}$ |
| AutoAgents | $16.13^{\pm 2.80}$ | $0.00^{\pm 0.00}$ | $0.00^{\pm 0.00}$ | $5.16^{\pm 0.95}$ |
| AgentSquare | $22.58^{\pm 2.15}$ | $15.72^{\pm 1.80}$ | $6.25^{\pm 1.50}$ | $16.34^{\pm 1.45}$ |
| AFlow | $10.75^{\pm 1.55}$ | $8.81^{\pm 1.25}$ | $4.08^{\pm 0.90}$ | $8.00^{\pm 0.85}$ |
| MaAS | $25.91^{\pm 1.90}$ | $22.01^{\pm 1.65}$ | $6.25^{\pm 1.20}$ | $20.69^{\pm 1.30}$ |
| **EvoMAS** | $30.11^{\pm 1.25}$ | $22.64^{\pm 1.15}$ | $8.61^{\pm 0.85}$ | $22.59^{\pm 0.90}$ |

*Table 4.* **Deployment-oriented cost (TCO).** We report total cost of ownership $\text{TCO@}N = C_{\text{train}} + N \cdot C_{\text{inf}}$ based on GPT-4o-mini pricing ($0.15/1M tokens).

| Method | Token Usage | | TCO (USD) ↓ | |
|---|---|---|---|---|
| | Tr. Tok. | Inf. Tok. | TCO@10 | TCO@100 |
| AFlow | 21.4M | 13.8M | 23.91 | 210.21 |
| EvoFlow | 13.7M | 10.6M | 17.96 | 161.06 |
| EvoMAS | 14.8M | 10.1M | 17.37 | 153.72 |

tent gains spanning arithmetic reasoning, program synthesis, and tool-augmented embodied tasks indicate that EvoMAS's role-level specialization and curriculum-guided evolution generalize robustly across heterogeneous domains rather than overfitting to any single benchmark.

**Cost Analysis.** We report the aggregated TCO across all six benchmarks using GPT-4o-mini pricing. Although EvoMAS incurs a slightly higher initial search cost than EvoFlow, it discovers streamlined architectures that significantly minimize per-run inference overhead. Consequently, EvoMAS achieves the lowest long-term operational costs ( TCO@100: **$153.72**), establishing it as the most economically viable solution for large-scale deployment.

The Pareto efficiency analysis in Figure 4 shows that EvoMAS occupies a favorable position on the Pareto frontier, achieving a superior performance–cost balance through its Cyber Creator, which implicitly steers evolution toward cost-effective structures via rule-based guidance rather than explicit cost terms in the objective. In particular, EvoMAS

attains accuracy on par with AFlow (gpt4o) while being roughly 3× cheaper, and its GPT-4o-mini variant further improves the accuracy–cost trade-off by trading marginal accuracy for substantially lower inference cost. These results validate a key insight: strategically selecting cost-effective executors (e.g., DeepSeekV3) for targeted optimization is more economical than exhaustive model exploration, making EvoMAS practical for resource-conscious deployments.

**Ablation Study.** Table 5 shows that each component of EvoMAS makes a substantial contribution to system performance across different task domains. Exploration and exploitation strategies are the most influential, yet task-dependent: removing exploitation caused the largest drop on MBPP ($-14.07\%$), while removing exploration caused the largest drop on MATH ($-12.23\%$), indicating that diverse search and local refinement play complementary roles. Notably, removing the Cyber Creator both reduced performance and increased computational costs (MATH: $+22.8\%$), highlighting its dual role in guiding evolution and controlling resource efficiency. These results demonstrate the synergistic effects of EvoMAS's components.

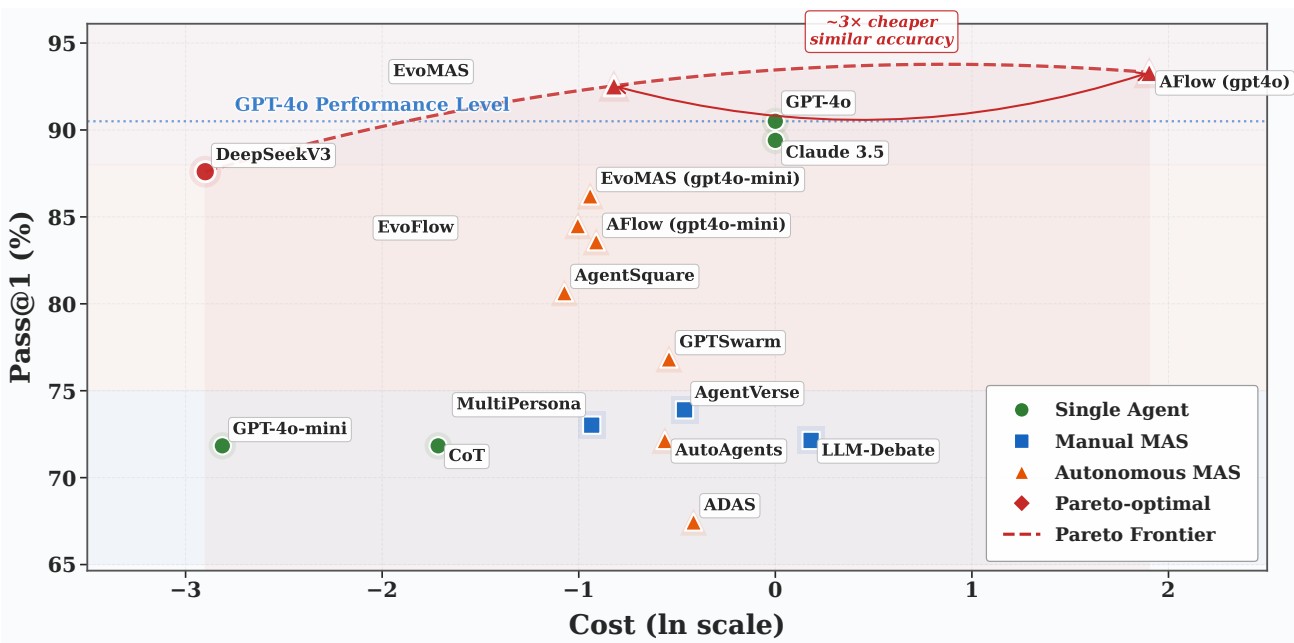

*Figure 4.* **Performance–Cost Trade-off of Agentic Workflow Methods.** We compare single-agent baselines, manually designed MAS, and autonomous MAS methods in terms of Pass@1 and inference cost in logarithmic scale. EvoMAS lies on the Pareto frontier, while its GPT-4o-mini variant further improves the accuracy-cost trade-off.

*Table 5.* **Ablation Study of EvoMAS.** We report task performance and average cost after removing each key component. Red values denote absolute performance drops.

| Variant | MBPP | | MATH | |
|---|---|---|---|---|
| | Pass@1 (%) | Cost ($) | Acc. (%) | Cost ($) |
| **Full EvoMAS** | **86.21** | 6.14 | **60.33** | 2.54 |
| w/o Explore | 76.83 ↓9.38 | 5.69 | 48.10 ↓12.23 | 1.94 |
| w/o Exploit. | 72.14 ↓14.07 | 6.24 | 50.17 ↓10.16 | 2.67 |
| w/o Creator | 78.52 ↓7.69 | 7.25 | 56.00 ↓4.33 | 3.12 |
| w/o Curric. | 83.57 ↓2.64 | 6.47 | 57.80 ↓2.53 | 2.77 |
| w/o Res. Lib. | 76.90 ↓9.31 | 5.22 | 55.37 ↓4.96 | 2.44 |

*Table 6.* **LLM Dependency Sensitivity.** Pass@1 (%) and standard deviation when replacing different LLM-dependent modules with weaker open-source models on MBPP. Qwen-122B, Qwen-35B, and Qwen-4B denote Qwen-3.5 series models of different scales.

| Module Replaced | Qwen-122B | Qwen-35B | Qwen-4B |
|---|---|---|---|
| **Full EvoMAS** | **86.2**±1.3 | – | – |
| Optimizer | – | 83.1±1.4 | 74.7±1.9 |
| Cyber Creator | – | 85.6±1.2 | 81.4±1.7 |
| Judge | – | 85.9±1.1 | 84.8±1.3 |

**Sensitivity to LLM-dependent modules.** We further evaluate the dependence of EvoMAS on LLM-based components by replacing the optimizer, Cyber Creator, and Judge with Qwen-3.5 models of different scales. As shown in Table 6, the optimizer is the most sensitive component: replacing it with Qwen-35B and Qwen-4B reduces Pass@1 from 86.2% to 83.1% and 74.7%, respectively. In contrast, replacing the Cyber Creator or Judge leads to much smaller degradation, even under the 4B setting. These results suggest that EvoMAS mainly relies on strong LLMs for generating effective structural variations, while its reflective control and difficulty-estimation components are relatively robust to weaker models.

The absence of the **Cyber Creator** not only reduced performance but also significantly increased costs (cost increased by 18%), demonstrating its dual value in guiding evolution direction and resource control. The removal of curriculum learning, and Evolution Resource Library also led to varying degrees of performance degradation, verifying the necessity of these components in knowledge accumulation, structural optimization, and agent role definition.

**Evolution of Character Generations.** As shown in Figure 6, the evolution of agent roles highlights how EvoMAS adapts its internal structure under task-driven pressures.

The system begins with a homogenous configuration of 10 **Actor** agents, reflecting an early-stage bias toward direct execution. However, this composition rapidly shifts as the system learns that complex mathematical reasoning requires more than isolated action. Roles such as **Planner** and **Critic** steadily increase, indicating a strategic pivot toward struc-

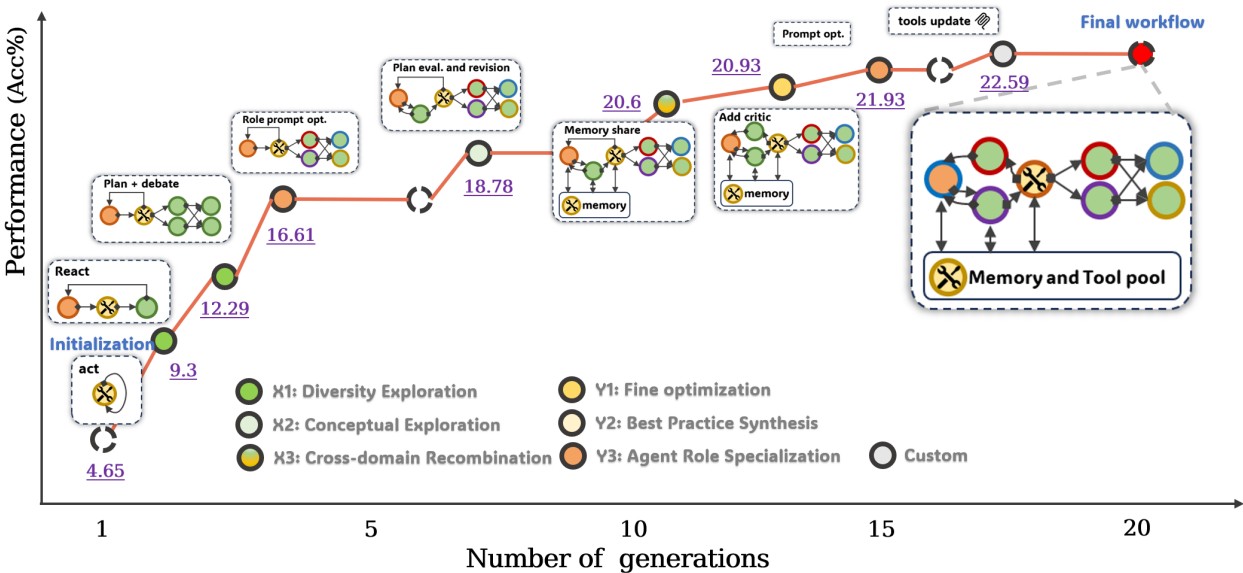

*Figure 5.* **Case Study: Evolutionary Trajectory of EvoMAS on GAIA.** Starting from a simple *act/ReAct* workflow, EvoMAS gradually evolves the system through planning, debate, role-prompt optimization, memory sharing, critic insertion, prompt refinement, and tool-pool updates. Colored markers indicate the evolutionary operators applied at each stage, while the curve shows the corresponding performance improvement across generations. The final workflow demonstrates how EvoMAS transforms an initially simple agent into a specialized multi-agent system with coordinated reasoning, memory, and tool-use capabilities.

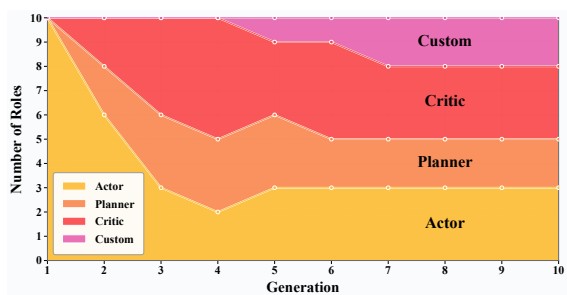

*Figure 6.* Dynamic Evolution of Agent Roles from Homogeneous Actors to Specialized Planners and Critics on the MATH Dataset.

tured problem decomposition and multi-perspective evaluation. From Generation 4 onward, Custom roles—introduced via the Cyber Creator—emerge in response to task-specific challenges that exceed the capabilities of standard roles. This dynamic reallocation demonstrates EvoMAS's capacity for structural self-optimization: rather than merely evolving workflows, it evolves the cognitive architecture behind them, ultimately converging toward a more balanced and reasoning-oriented agent ensemble.

**Case Study.**  Figure 5 illustrates how EvoMAS evolved from a simple template to a high-performance multi-agent workflow. The system initially contained only a simple node, then underwent step-by-step modifications in each generation, such as adding nodes and optimizing prompts. This process embodies the collaborative dynamics of the

exploration-exploitation-reflection mechanism in the Evo-MAS framework: in the early stages, structural diversity stimulates potential solution spaces; in the middle stages, guided strategies focus on performance-critical paths; in the later stages, rule injection and reflection updates accelerate evolutionary convergence, demonstrating powerful self-optimization and adaptive capabilities.

## 6. Conclusion

We present EvoMAS, a biologically-inspired framework for automatically evolving multi-agent systems. By integrating role-level evolution, dynamic and diverse evolutionary strategies, and curriculum-guided learning, EvoMAS addresses the rigidity and limited cross-task generalization of existing automated designs, while the Cyber Creator steers the search through rule-based governance and periodic reflective updates. Across diverse benchmarks, Evo-MAS consistently outperforms strong baselines and prior SOTA methods while maintaining superior cost-efficiency. Nevertheless, its effectiveness still hinges on a sufficiently capable optimizer LLM for generating high-quality structural variations and on LLM-based judgment for selection and difficulty estimation, which may introduce bias or instability under weaker and noisier models. More broadly, we view learning and refining heuristics in the loop—rather than hand-crafting them—as an increasingly central paradigm for building scalable, self-improving LLM agents, and we hope EvoMAS offers a concrete step toward that future.

## Impact Statement

This work aims to improve the scalability, efficiency, and adaptability of LLM-based multi-agent systems by automating workflow design. It may lower the engineering barrier for deploying agentic systems in domains such as software engineering, data analysis, and embodied decision-making. At the same time, more automated multi-agent systems may raise concerns about reliability, transparency, and misuse. EvoMAS partially mitigates these risks through structured evolution, explicit role specialization, and cost-aware analysis, but responsible deployment still requires robustness evaluation, safeguards, and human oversight.

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

# A. Supplementary Results

## A.1. Stability Trends with Varying Agent Width and Depth

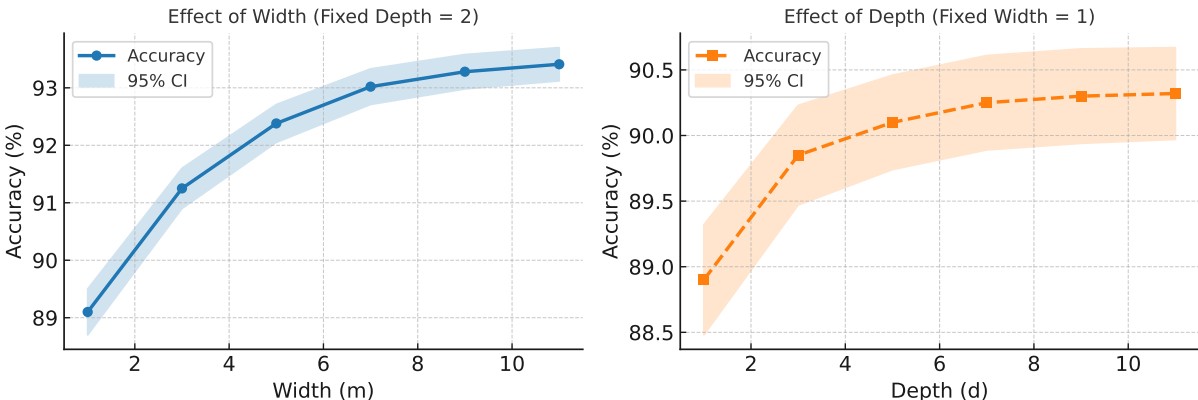

*Figure 7.* Accuracy and 95% confidence intervals (CI) across varying widths (left, fixed depth = 2) and depths (right, fixed width = 1). Shaded regions indicate the CI range.

Figure 7 presents the accuracy trends and stability (95% CI) under two configurations: increasing agent width with fixed depth (left) and increasing agent depth with fixed width (right).

**Width Analysis:** As width increases from $m = 1$ to $m = 11$, accuracy improves steadily while the confidence interval narrows. This suggests that with more agents participating, the collective decision becomes more stable due to diversity and redundancy. However, after $m = 7$, the marginal gain in CI reduction diminishes, indicating saturation in collaborative benefits.

**Depth Analysis:** Increasing depth from $d = 1$ to $d = 11$ initially enhances stability (from $d = 1$ to $d = 3$), reflecting the value of multi-step reasoning or negotiation among agents. However, beyond $d = 5$, the CI plateaus and eventually shows negligible improvement. This implies that deeper structures may suffer from information distortion or diminishing returns due to accumulated reasoning noise.

**Emergent Principle:** A key insight from these results is the *non-linear convergence of stability*:

- Increasing **width** promotes stability via agent diversity and ensemble averaging, but its benefit saturates as inter-agent redundancy increases.

- Increasing **depth** initially enhances agreement through reasoning chains, but deeper layers may introduce instability from noise accumulation or misalignment.

Hence, optimal stability in multi-agent systems may require a balanced coordination strategy that avoids both shallow reasoning and excessive architectural complexity.

## A.2. Role Co-Occurrence in Workflow Graphs

To better understand the internal organization of evolved multi-agent workflows, we analyze the structural co-occurrence patterns of different agent roles within the workflow graphs. Specifically, we construct a role co-occurrence matrix where each entry $(i, j)$ represents the number of edges from role $i$ to role $j$ across all workflows evolved on the MATH dataset.

Figure 8 presents the resulting heatmap, providing a graphical summary of role connectivity patterns.

## A.3. Training Configuration and Cost Transparency

To ensure reproducibility and transparency, we provide the full training configuration of EvoMAS together with a detailed computational resource and cost analysis. Unlike continuous training paradigms that require massive GPU compute,

- **Planner** nodes frequently connect with both **Actor** and **Critic** roles, reflecting their central role in coordinating task decomposition and quality control.
- **Critic–Planner** and **Planner–Actor** links dominate, forming the backbone of a reflective planning–execution loop.
- **Custom** roles show dispersed connections to all other types, highlighting their flexible, late-stage integration into evolved workflows.
- Diagonal values indicate intra-role cooperation (e.g., Critic $\rightarrow$ Critic), commonly seen in complex reasoning chains.

These co-occurrence patterns reveal EvoMAS's tendency to converge on a modular architecture in which planning, acting, and evaluating are handled by specialized but tightly coupled agent roles.

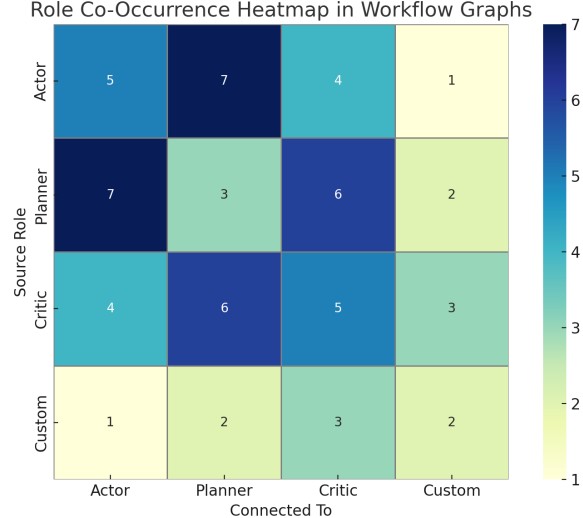

*Figure 8.* Role Co-Occurrence Heatmap in Workflow Graphs. Each cell $(i, j)$ indicates the number of directed edges from role $i$ to role $j$ across evolved workflows.

EvoMAS adopts a one-time structural search approach using efficient LLM APIs. This design choice keeps the computational overhead minimal while achieving strong generalization capabilities.

*Table 7.* **Training Phase Cost.** Token usage and cost statistics of EvoMAS training experiments (Pricing: GPT-4o-mini at $0.15/1M tokens).

| Task | Dataset | Training Samples | Iterations | Total Candidates | Token Usage (M) | Cost (USD) |
|---|---|---|---|---|---|---|
| Code Generation | HumanEval | 33 | 20 | 337 | 2.17M | $0.33 |
| Math Reasoning | GSM8K | 264 | 20 | 345 | 1.76M | $0.26 |
| Tool Usage | GAIA | 94 | 20 | 413 | 2.49M | $0.37 |
| | | | | **Total (All Benchmarks)** | **14.8M** | **$2.22** |

All experiments were conducted using the OpenAI GPT-4o-mini API, eliminating the need for local GPU clusters. The cost analysis presented in Table 7 is calculated based on the standard pricing rate of **$0.15 per 1 million tokens**.

**Extreme Cost Efficiency.** As summarized in Table 7, the aggregated token consumption across all experimental tasks (including Code Generation, Math Reasoning, and Tool Usage) was approximately **14.8 million tokens**. Remarkably, due to the efficiency of the underlying model and our structured search strategy, the **total financial cost for the entire training phase was merely $2.22**. This low barrier to entry demonstrates that EvoMAS democratizes the automated design of complex agentic systems, making state-of-the-art workflow evolution accessible even with limited computational budgets.

### A.4. Sensitivity Analysis of Reflection Interval

In the EvoMAS framework, the reflection interval $K$ serves as a critical hyperparameter regulating the trade-off between evolutionary convergence speed and computational resource efficiency. As derived in our theoretical analysis (refer to Eq. 27), the performance gain introduced by the Cyber Creator is proportional to the frequency of meta-updates ($\propto 1/K$). This suggests that smaller $K$ values (higher reflection frequency) theoretically facilitate faster correction of the policy distribution. To empirically validate this and identify the optimal configuration, we conducted a sensitivity analysis on the reflection interval $K \in \{2, 5, 10, 20\}$.

As presented in Table 8, the setting of $K = 5$ achieves the optimal balance among accuracy (66.9%), convergence speed, and computational cost. We observe distinct behavioral regimes based on the frequency of reflection:

▶ **Over-correction** ($K = 2$)**:** While frequent updates provide tight theoretical bounds, an excessively short interval leads to instability in the rule set and high overhead ($2.94), resulting in a suboptimal invalid structure ratio (17.5%).

▶ **Under-adaptation** ($K = 20$)**:** Infrequent reflection fails to prune ineffective strategies in a timely manner, significantly

*Table 8.* Impact of Reflection Interval $K$ on Performance, Convergence, and Cost Efficiency. All experiments were conducted on the MATH dataset with a fixed population size.

| Interval ($K$) | Final Acc. (%) | Conv. Gen. | Tokens (M) | Refl. Calls | Cost ($) | Invalid Ratio (%) |
|---|---|---|---|---|---|---|
| 2 | 63.4 | 0.8 | 382 | 50 | 2.94 | 17.5 |
| **5** | **66.9** | **0.6** | **241** | **20** | **1.98** | **6.2** |
| 10 | 65.1 | 0.7 | 311 | 10 | 1.82 | 9.6 |
| 20 | 61.7 | 1.1 | 451 | 5 | 1.65 | 21.8 |

delaying convergence (1.1 generations) and degrading final accuracy to 61.7%.

**The Cost-Efficiency Paradox.** A counter-intuitive phenomenon observed is that maintaining a moderate reflection frequency ($K = 5$) is more cost-effective than removing the reflection mechanism entirely. As noted in our main discussion, removing the Cyber Creator (conceptually equivalent to $K \to \infty$) increases the total cost by approximately 18%. This occurs because the absence of timely meta-strategy guidance allows the population to propagate a large number of low-quality or invalid structures. The token consumption required for these inefficient agents to perform downstream inference tasks far exceeds the marginal overhead introduced by the Cyber Creator's reflective calls.

Consequently, the choice of $K = 5$ is not merely heuristic but represents an empirically justified equilibrium point that maximizes the "Return on Intelligence" for the meta-control system.

# B. EvoMAS Framework Details

## B.1. Overall Algorithm Description

EvoMAS employs a hierarchical evolutionary framework with curriculum learning, operating across three dimensions: role specialization, strategy selection, and curriculum progression as shown in Fig. 9.

**Core Evolution Process:** Each generation follows a variation-selection-reflection cycle using six biologically-inspired operators—three for exploration (diversity expansion, conceptual recombination, cross-domain hybridization) and three for exploitation (fine optimization, best practice synthesis, role specialization). LLM-based selection evaluates candidates on performance, structure, and rule compliance rather than scalar fitness alone. Every $K$ generations, the Cyber Creator performs meta-reflection to update rules and strategy distributions.

**Curriculum-Guided Evolution:** Tasks are partitioned by difficulty levels, with workflows evolved sequentially from simple to complex. Cross-stage stability constraints prevent catastrophic forgetting through cumulative performance evaluation $J_k(G) = \frac{1}{k} \sum_{i=1}^{k} f(G, T_i)$, ensuring evolved workflows maintain competence across all previous difficulty levels.

---

**Algorithm 1** EvoMAS: Core Evolutionary Algorithm

---

**Require:** Task distribution $T$, population size $N$, generations $G$, reflection interval $K$
**Ensure:** Best evolved workflow $G^*$
 1: INITIALIZE population $P_0 = \{G_1, G_2, \ldots, G_N\}$ via RAG retrieval
 2: INITIALIZE rule set $R_0$, strategy distribution $A_0$
 3: INITIALIZE evolution resource library (rule pool, gene pool)
 4: **for** $t = 1 \to G$ **do**
 5:    // **VARIATION STAGE**
 6:    $P_t^{\text{VAR}} \leftarrow \emptyset$
 7:    **for** each parent $G \in P_{t-1}$ **do**
 8:       SAMPLE strategy $a \sim A_{t-1}$ {Six biological operators}
 9:       GENERATE offspring $G' \sim a(G, R_{t-1})$
10:       $P_t^{\text{VAR}} \leftarrow P_t^{\text{VAR}} \cup \{G'\}$
11:    **end for**
12:    // **SELECTION STAGE**
13:    **for** each candidate $G \in P_{t-1} \cup P_t^{\text{VAR}}$ **do**
14:       COMPUTE fitness $F(G, T) = \mathbb{E}_{x \sim T}[f(G, x)]$
15:       SERIALIZE $G$ with performance, structure, and rule compliance
16:    **end for**
17:    $P_t \leftarrow \text{LLMSELECT}(P_{t-1} \cup P_t^{\text{VAR}}, F, R_{t-1})$ {Preference-based}
18:    // **REFLECTION STAGE (EVERY $K$ GENERATIONS)**
19:    **if** $t \bmod K = 0$ **then**
20:       UPDATE historical log $H_t \leftarrow H_{t-1} \cup \{(G_i, F_i, A_i)\}_{i \leq t}$
21:       $R_t \leftarrow \text{CYBERCREATOR}(R_{t-1}, H_t)$ {Rule update}
22:       $A_t \leftarrow \text{STRATEGYUPDATE}(A_{t-1}, H_t)$ {Strategy adaptation}
23:       UPDATE evolution resource library with successful patterns
24:    **else**
25:       $R_t \leftarrow R_{t-1}, A_t \leftarrow A_{t-1}$
26:    **end if**
27: **end for**
28:
29: **return** $G^* = \arg\max_{G \in \bigcup_{i=1}^{G} P_i} F(G, T, R)$

---

## B.2. Mathematical Formalization of Evolutionary Operators

To establish a rigorous theoretical foundation for EvoMAS, we model the agentic workflow generation as a stochastic optimization problem over a discrete graph space. Let $\mathbb{G}$ denote the space of all admissible directed acyclic graphs (DAGs), where each workflow $G \in \mathbb{G}$ is defined as a tuple $G = (V, E, \Phi)$. Here, $V$ represents the set of agent nodes, $E \subseteq V \times V$

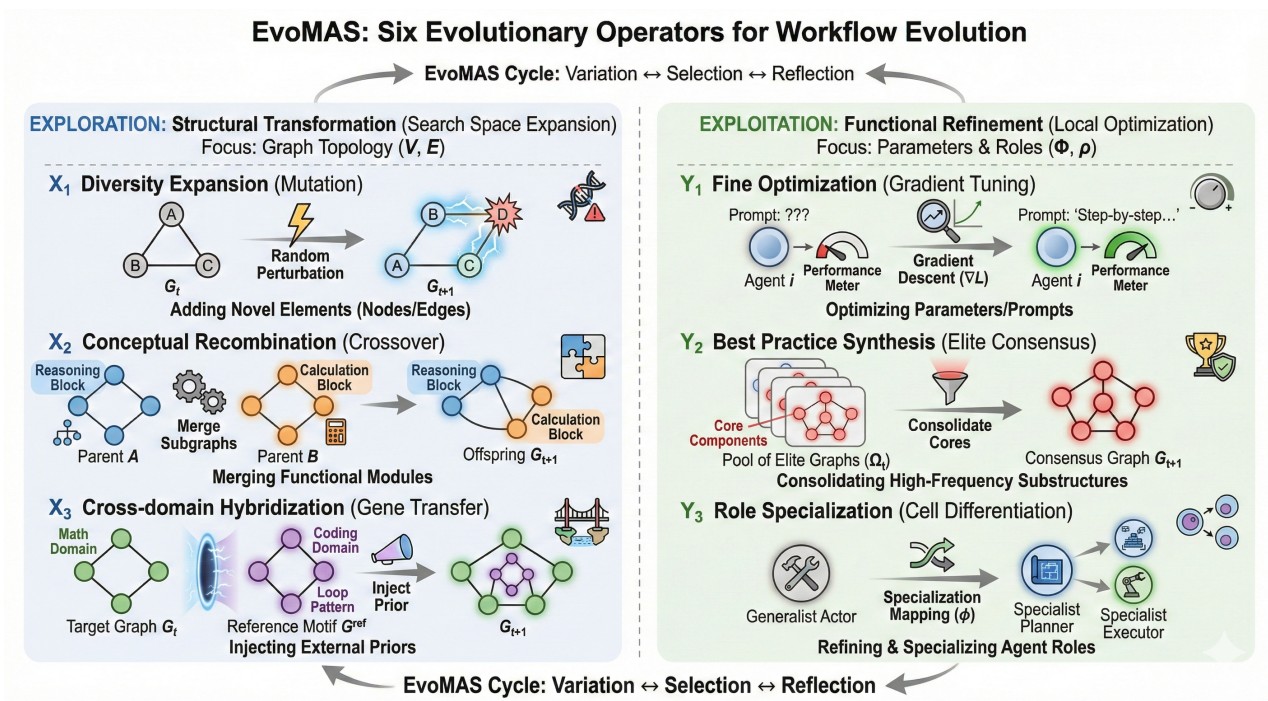

*Figure 9.* illustration of six biologically-inspired operators.

denotes the communication topology, and $\Phi = \{\phi_v\}_{v \in V}$ encompasses the parameter space (e.g., prompts, roles, tools) associated with each node. The evolutionary process is governed by a transition operator $\mathcal{T} : \mathbb{G} \to \mathbb{G}$, which maps the current population state to the next via a composition of **Exploration** ($\mathcal{X}$) and **Exploitation** ($\mathcal{Y}$) functionals:

$$G_{t+1} \sim \mathcal{T}(G_t; \mathcal{X}, \mathcal{Y}) \tag{12}$$

### B.2.1. EXPLORATION OPERATORS: TOPOLOGICAL STOCHASTICITY

Exploration operators are designed to induce topological diversity, effectively expanding the search envelope within $\mathbb{G}$ to escape local optima.

$\mathcal{X}_1$: **Diversity Expansion (Stochastic Perturbation).** This operator introduces entropy into the system by applying a noise function $\delta$ to the graph topology. It mimics *genetic mutation* by stochastically injecting or pruning nodes/edges from a candidate pool $\mathcal{V}_{\text{new}}$:

$$G_{t+1} = (V_t \Delta V_\delta, \ E_t \Delta E_\delta), \quad \text{where } V_\delta \sim \text{Bernoulli}(\mathcal{V}_{\text{new}}). \tag{13}$$

Here, $\Delta$ denotes the symmetric difference, ensuring reversible structural variations.

$\mathcal{X}_2$: **Conceptual Recombination (Subgraph Crossover).** Analogous to *sexual recombination*, this operator constructs a new topology by merging functional subgraphs from two parent workflows $G_a, G_b \in \mathbb{G}$. Let $\mathcal{S}(G)$ be a subgraph extraction function; the recombination is defined as the graph union of disjoint functional modules:

$$G_{t+1} = \mathcal{S}(G_a) \cup \mathcal{S}(G_b) \cup E_{\text{bridge}}, \tag{14}$$

where $E_{\text{bridge}}$ represents the newly established edges connecting the heterogeneous modules.

$\mathcal{X}_3$: **Cross-domain Hybridization (Structural Injection).** Inspired by *horizontal gene transfer*, this operator projects a high-performing motif from a reference domain (source graph $G^{\text{ref}}$) onto the target graph $G_t$. We define this as a convex combination in the structural space, controlled by a hybridization factor $\alpha \in [0, 1]$:

$$G_{t+1} = G_t \oplus_\alpha \psi(G^{\text{ref}}), \tag{15}$$

where $\psi$ is an isomorphism mapping that aligns the reference motif with the target context, and $\oplus_\alpha$ denotes the probabilistic injection of structural priors.

### B.2.2. EXPLOITATION OPERATORS: MANIFOLD REFINEMENT

Exploitation operators focus on local optimization within the neighborhood of high-fitness structures, aiming to maximize the objective function $J(G)$ (e.g., accuracy).

$\mathcal{Y}_1$**: Fine Optimization (Parametric Gradient Descent).** This operator performs local search in the continuous parameter space $\Phi$ while keeping the topology $(V, E)$ fixed. Assuming a generalized loss landscape $L$, the update follows a pseudo-gradient descent trajectory:

$$\Phi_{t+1} = \Phi_t - \eta \cdot \hat{\nabla}_\Phi \mathbb{E}[L(G_t)], \tag{16}$$

where $\eta$ is the learning rate and $\hat{\nabla}$ represents a gradient estimator (e.g., text-based feedback or numerical approximation) for non-differentiable LLM components.

$\mathcal{Y}_2$**: Best Practice Synthesis (Elite Consensus).** Unlike simple selection, this operator synthesizes a robust workflow by aggregating the "common core" of the elite population $\Omega_t \subset \mathbb{G}$. It can be viewed as finding the structural centroid:

$$G_{t+1} = \bigcup_{G_k \in \Omega_t} (G_k \cap \mathcal{K}_{\text{core}}), \tag{17}$$

where $\mathcal{K}_{\text{core}}$ denotes the high-frequency substructures observed across top-performing agents, effectively filtering out noise while preserving effective interaction patterns.

$\mathcal{Y}_3$**: Role Specialization (Function Differentiation).** Modeled after *cellular differentiation*, this operator refines the semantic role $\rho_v$ of a specific node $v$. It applies a specialization mapping $\mathcal{M}$ that transforms a generalist role into a specialist one to minimize task ambiguity:

$$\rho_v^{(t+1)} = \mathcal{M}(\rho_v^{(t)} | \mathcal{H}_{\text{task}}), \tag{18}$$

where $\mathcal{H}_{\text{task}}$ represents the hierarchical decomposition of the current task constraints.

---

**Algorithm 2** EvoMAS: Curriculum-Guided Evolution

---

**Input:** task distribution $T$, difficulty layers $n$, competence threshold $\tau$
**Output:** multi-stage evolved population $P^{final}$
**Task Difficulty Partitioning**
Partition $T = \{T_1, T_2, \ldots, T_n\}$ such that $d(T_1) < d(T_2) < \cdots < d(T_n)$
Initialize population $P^{(0)} \leftarrow$ INITIALIZE()
$G_0^{best} \leftarrow$ null
**for** $k = 1$ **to** $n$ **do**
    **Sequential Evolution on Stage** $k$
    $(G_k^{best}, P^{(k)}) \leftarrow$ EVOMAS-EVOLUTION$(T_k, P^{(k-1)})$
    **Stability Constraint Check**
    **if** $k > 1$ **then**
        $J_k(G_k^{best}) \leftarrow \frac{1}{k} \sum_{i=1}^k f(G_k^{best}, T_i)$
        $J_k(G_{k-1}^{best}) \leftarrow \frac{1}{k} \sum_{i=1}^k f(G_{k-1}^{best}, T_i)$
        **if** $J_k(G_k^{best}) \leq J_k(G_{k-1}^{best})$ **then**
            **reject** $G_k^{best}$, continue evolution on $T_k$
        **end if**
    **end if**
    **Competence Check**
    $perf_k \leftarrow F(G_k^{best}, T_k)$
    **if** $perf_k < \tau$ **then**
        Continue evolution on $T_k$ until $perf_k \geq \tau$
    **end if**
    **Knowledge Transfer and Stage Transition**
    Update evolution resource library
    $P^{(k)} \leftarrow$ select elite population for next stage
**end for**
**Return** $P^{final} \leftarrow P^{(n)}$

---

## B.3. "Cyber Creator" Implementation

The "Cyber Creator" in EvoMAS serves as a meta-controller that adaptively regulates the evolutionary process via external rule guidance and reflection-based strategy revision. This appendix provides implementation details, including the format of rules, prompting strategies, and concrete examples of its operation.

*Table 9.* Summary of Evolutionary Strategies in EvoMAS and Their Biological Analogies

| | Strategy Type | # Parents | Biological Analogy | Description |
|---|---|---|---|---|
| $X_1$ | Explor. | Multiple | *Adaptive Radiation* | Generates structurally diverse workflows by varying topology and role composition, analogous to species diversification into new ecological niches. |
| $X_2$ | Explor. | Multiple | *Sexual Recombination* | Preserves core functional ideas while varying implementations through genetic crossover to promote innovation. |
| $X_3$ | Explor. | Cross-domain | *Horizontal Gene Transfer* | Combines substructures from workflows in different domains, mimicking bacterial exchange of genetic material across species. |
| $Y_1$ | Exploit. | Single | *Microevolution* | Performs incremental local refinements through natural selection pressure on specific performance bottlenecks. |
| $Y_2$ | Exploit. | Multiple | *Selective Breeding* | Merges effective components from multiple workflows, analogous to artificial selection for desired traits. |
| $Y_3$ | Exploit. | Single | *Cellular Differentiation* | Enhances specialization by refining agent roles, similar to how cells develop distinct functions in multicellular organisms. |
| Custom | Mixed | Variable | *Epigenetic Regulation* | Dynamically defined by the *Cyber Creator* through environmental feedback, analogous to gene expression modification without altering DNA sequence. |

## B.4. Other Implementation Details

### B.4.1. RULE REPRESENTATION AND GENERATION

To bridge the gap between abstract constraints and actionable evolutionary guidance, we formalize each evolution rule as a semantic triplet $r_i = (c_i, w_i, d_i)$. Here, $c_i$ denotes the **Condition** (a logic-based or natural language constraint, e.g., num_planners$(G) \leq 1$); $w_i \in$ {High, Medium, Low} represents the **Weight** or priority; and $d_i$ provides a natural language **Description** of the rule's intent (e.g., "Encourage shallow structures to reduce latency"). During the variation and selection phases, these triplets are injected into the LLM context as soft constraints, biasing the stochastic generation towards structurally sound regions.

**Automated Rule Synthesis.** EvoMAS acts as an *Evolution Overseer*, querying an LLM to deduce new rules from historical data. The prompt template includes a summary of parental performance and a history log $H_t = \{(G_k, F_k, \pi_k)\}$, requesting the synthesis of rules that reduce redundancy or enhance generalization. For instance, given a history where high-performing agents utilized fewer than 4 actors, the LLM might output a JSON object: {"condition": "num_agents(G)<=4", "weight": "Medium", "description": "Limit agent count for efficiency"}. Similarly, custom strategies are synthesized by identifying recurring bottlenecks; for example, observing nested Critic→Planner cycles triggers a *Feedback Loop Rewriter* strategy that flattens nested interactions into composite roles.

## B.5. Case Study: Evolutionary Trajectory in Code Debugging

We illustrate the efficacy of EvoMAS through a concrete Code Debugging scenario. Table 10 details the trajectory where performance evolved from 0.62 to 0.80 over three generations.

*Table 10.* **Evolutionary Trajectory.** Step-by-step refinement of the workflow structure and rule set. **Key:** `P`=Planner, `A`=Actor, `C`=Critic. Notations like $\{\cdot\}$ denote parallel execution.

| Gen | Phase | Dominant Operator & Structure | Rule Dynamics $(\triangle R)$ | Score |
|---|---|---|---|---|
| 0 | *Init* | **Template Retrieval** 
 $G_0 : \texttt{P} \to \texttt{A} \to \texttt{C}$ | $R_0$: "Depth $\leq 5$" 
 (Initial Constraint) | 0.62 |
| 1 | *Explore* | $\mathcal{X}_1$ **(Diversity)** $\to$ Parallelism 
 $G_1 : \texttt{P} \to \{\texttt{A}_1, \texttt{A}_2\} \to \texttt{C}$ | $R_1$: "Actor Count $\geq 2$" (New) | 0.70 |
| 2 | *Hybrid* | $\mathcal{X}_3$ **(Cross-domain)** $+ \mathcal{Y}_2$ **(Synth)** 
 $G_2 : \texttt{P} \to \{\texttt{A}_1, \texttt{A}_2^{\text{tools}}\} \to \texttt{C}$ | $R_1$: Weight $\uparrow$ (High) 
 $R_0$: Pruned (Obsolete) | 0.77 |
| 3 | *Refine* | $\mathcal{Y}_1$ **(Fine-Opt)** $\to$ Prompt Tuning 
 $G_3 : \texttt{P} \to \{\texttt{A}_1, \texttt{A}_2^{\text{tuned}}\} \to \texttt{C}$ | Stability Check $\checkmark$ | **0.80** |

### B.5.1. EVOLUTION RESOURCE LIBRARY VIA GRAPH RAG

To prevent catastrophic forgetting and facilitate knowledge transfer, EvoMAS maintains an **Evolution Resource Library** managed by a Graph Retrieval-Augmented Generation (Graph RAG) system based on LightRAG (Guo et al., 2025). This system integrates structural graph data with vector embeddings to manage two core repositories:

**Rule Pool & Gene Pool Dynamics.** The *Rule Pool* stores evolutionary guidance, categorized into *Variation Rules* (guiding mutations) and *Selection Rules* (filtering candidates). Rules form a citation graph where edges denote relationships such as conflict, entailment, or derivation. Simultaneously, the *Gene Pool* archives high-performing structural motifs $g_i = (s_i, p_i, u_i)$, where $s_i$ is the subgraph structure, $p_i$ the performance metric, and $u_i$ usage statistics. The system performs periodic maintenance operations: *Gene Merging* consolidates functionally isomorphic subgraphs to reduce redundancy; *Gene Elimination* prunes underutilized components; and *Gene Analysis* extracts design principles from frequent motifs.

**Dual-Level Retrieval Mechanism.** LightRAG enables a dual-granularity retrieval paradigm essential for evolutionary search. *Low-level retrieval* focuses on precise entity matching (e.g., specific agent roles), while *high-level retrieval* captures broad structural themes (e.g., "hierarchical planning patterns") via multi-hop traversals. An incremental update mechanism allows new evolutionary findings to be appended to the knowledge graph without costly re-indexing, ensuring the system adapts in real-time.

## B.6. Complexity Analysis and Search Space Pruning

The design space of agentic workflows $\mathbb{G}$ suffers from combinatorial explosion; for a graph with $N$ nodes and $K$ role types, the complexity scales as $O(K^N \cdot 2^{N^2})$, rendering naive search intractable. EvoMAS addresses this via two theoretical constraints derived from our design assumptions: (1) **Topological Sparsity** ($\mathcal{H}_3$): We impose hard constraints on maximum node degree and graph depth, restricting the search to a sparse subspace $\mathbb{G}_{sparse} \subset \mathbb{G}$. (2) **Functional Modularity** ($\mathcal{H}_1$): By evolving functional modules via Operator $\mathcal{X}_2$ rather than individual edges, we coarsen the optimization granularity from edge-level to block-level. These constraints effectively reduce the effective search dimension, allowing EvoMAS to converge to high-quality solutions within limited generations ($T \approx 20$) despite the vast theoretical bounds.

## C. Theoretical Analysis of EvoMAS Framework

In this section, we provide a formal analysis of the EvoMAS framework to elucidate its performance advantages. We model the system as a stochastic optimization process over a discrete graph space and analyze its efficiency from three perspectives: (1) search-space reduction via structured operators, (2) asymptotic convergence properties, and (3) acceleration through meta-Bayesian optimization.

*Table 11.* Key Notation for Theoretical Analysis

| Symbol | Description |
|---|---|
| $\mathbb{G}$ | The universal space of all possible agentic workflow graphs |
| $G^* \in \mathbb{G}$ | The global optimal workflow with maximum fitness $J(G^*)$ |
| $\mathcal{T}_\pi(G)$ | Transition operator under policy $\pi$, mapping $G_t \rightarrow G_{t+1}$ |
| $\Omega_R \subset \mathbb{G}$ | Constrained subspace defined by rule set $R$ |
| $\mathcal{X}, \mathcal{Y}$ | Exploration and Exploitation operator sets |
| $H_t$ | Evolutionary history up to generation $t$ |
| $\Delta J_t$ | Fitness improvement at generation $t$, $J(G_{t+1}) - J(G_t)$ |

### C.1. Search Efficiency via Structured Operators

The primary challenge in evolving agentic workflows is the combinatorial explosion of the graph space $\mathbb{G}$. For a directed graph with $N = |V|$ nodes, the adjacency matrix admits $2^{N^2}$ distinct configurations, and with $K$ role types per node a naive search has complexity $O\big(K^{|V|} \cdot 2^{|V|^2}\big)$. EvoMAS mitigates this through **structured variation**: every operator induces only a *local* move whose graph edit distance is bounded.

**Proposition C.1** (Effective Search Space Reduction). *Let $\mathbb{G}_{reach}(G) \subseteq \mathbb{G}$ be the set of graphs reachable from $G$ via a single application of operators $\mathcal{X} \cup \mathcal{Y}$. Under the bounded graph edit distance $d(G, G') \leq \delta$ imposed by our operators, the neighborhood size (branching factor) is polynomial in the graph size $|V|$:*

$$|\mathbb{G}_{reach}(G)| = O\big(|V|^2 + |V| \cdot |\Phi|\big), \tag{19}$$

*where $\Phi$ is the parameter space per node.*

*Proof.* Since $\mathcal{O} = \mathcal{X} \cup \mathcal{Y}$ acts on disjoint search manifolds (topology vs. parameters), we bound each contribution separately and sum.

**Exploration $\mathcal{X}$ (topological moves).** Node insertion/deletion chooses one of $|V|$ candidate slots, giving $O(|V|)$ options. Edge rewiring selects an unordered vertex pair, so its count is the binomial coefficient

$$\binom{|V|}{2} = \frac{|V|\,(|V|-1)}{2} = O(|V|^2). \tag{20}$$

Subgraph crossover ($\mathcal{X}_2$) is defined by cut-point selection on edges and is therefore also upper-bounded by $O(|V|^2)$. Summing the two move types,

$$|\mathcal{X}(G)| = \underbrace{O(|V|)}_{\text{nodes}} + \underbrace{O(|V|^2)}_{\text{edges}} \leq c_1 |V|^2. \tag{21}$$

**Exploitation $\mathcal{Y}$ (parametric moves).** These keep $(V, E)$ fixed and perturb the parameters $\phi_v$ of each node independently. With at most $O(|\Phi|)$ candidate states per node,

$$|\mathcal{Y}(G)| = \sum_{v \in V} O(|\Phi|) = O\big(|V| \cdot |\Phi|\big). \tag{22}$$

**Combination.** Adding (21) and (22),

$$|\mathbb{G}_{reach}(G)| = |\mathcal{X}(G)| + |\mathcal{Y}(G)| = O\big(|V|^2 + |V| \cdot |\Phi|\big). \tag{23}$$

Relative to brute-force enumeration, the branching factor shrinks super-exponentially:

$$\frac{|\mathbb{G}_{reach}(G)|}{|\mathbb{G}|} = \frac{O(|V|^2 + |V||\Phi|)}{2^{|V|^2}} \xrightarrow{|V| \to \infty} 0, \tag{24}$$

confirming that structured operators confine each step to a polynomial-sized local manifold. $\square$

## C.2. Convergence Analysis under Uncertainty

In realistic settings the fitness $J(G)$, derived from LLM feedback or stochastic benchmarks, is non-deterministic. We therefore analyze convergence to an $\epsilon$-**optimal set** under bounded evaluation noise. Let $\mathcal{F}_t$ denote the filtration generated by the evolutionary history $H_t$.

**Assumption C.2** ($\delta$-Bounded Evaluation Noise)**.** Let $\hat{J}(G)$ be the observed fitness and $J_{true}(G)$ the latent true fitness. The evaluation noise is uniformly bounded by $\delta \geq 0$:

$$|\hat{J}(G) - J_{true}(G)| \leq \delta, \quad \forall G \in \mathbb{G}. \tag{25}$$

**Assumption C.3** (Uniform Reachability)**.** For any $G, G' \in \mathbb{G}$, the exploration operators $\mathcal{X}$ assign a non-zero probability to a transition path of finite length $k$, uniformly over the starting state:

$$P(G_{t+k} = G' \mid G_t = G) \geq \gamma > 0. \tag{26}$$

**Theorem C.4** (Convergence to $\epsilon$-Optimal Set)**.** *Let* $S_\epsilon = \{G \mid J_{true}(G) \geq J_{true}(G^*) - \epsilon\}$ *with* $\epsilon > 2\delta$*. Under Assumptions C.2–C.3 and Elitist Selection on* $\hat{J}$*, the system visits* $S_\epsilon$ *almost surely:*

$$\mathbb{P}\left(\lim_{t \to \infty} \min_{\tau \leq t} \text{dist}(G_\tau, S_\epsilon) = 0\right) = 1. \tag{27}$$

*Proof.* **Step 1 (Reachability of $G^*$).** By Assumption C.3, from any state $G_t$ the candidate $G^*$ is generated within $k$ steps with probability at least $\gamma$.

**Step 2 (Survival under noise).** If $G^*$ is generated it must survive elitist selection against any $G' \notin S_\epsilon$. Chaining (25) with the definition of $S_\epsilon$ and the gap condition $\epsilon > 2\delta$:

$$\begin{aligned}
\hat{J}(G') &\leq J_{true}(G') + \delta & \text{(noise bound)} \\
&< \left(J_{true}(G^*) - \epsilon\right) + \delta & (G' \notin S_\epsilon) \\
&= J_{true}(G^*) - (\epsilon - \delta) \\
&< J_{true}(G^*) - \delta & (\epsilon > 2\delta) \\
&\leq \hat{J}(G^*). & \text{(noise bound)}
\end{aligned} \tag{28}$$

Hence $\hat{J}(G^*) > \hat{J}(G')$: noise of magnitude $\delta$ can never make an out-of-$S_\epsilon$ candidate outrank $G^*$.

**Step 3 (Almost-sure hitting).** Partition time into disjoint windows of length $k$. Because the bound (26) is uniform over the starting state, for each window $i$,

$$P(\text{miss}_i \mid \mathcal{F}_{(i-1)k}) \leq 1 - \gamma. \tag{29}$$

Applying the tower property over $m$ consecutive windows,

$$\begin{aligned}
P\left(\bigcap_{i=1}^m \text{miss}_i\right) &= \mathbb{E}\left[\prod_{i=1}^m P(\text{miss}_i \mid \mathcal{F}_{(i-1)k})\right] \\
&\leq (1 - \gamma)^m.
\end{aligned} \tag{30}$$

Taking $m \to \infty$,

$$P(\text{never reach } S_\epsilon) = \lim_{m \to \infty} (1 - \gamma)^m = 0. \tag{31}$$

Therefore $S_\epsilon$ is reached with probability 1, and by Step 2 elitism keeps the best-so-far solution inside $S_\epsilon$ thereafter, establishing (27). $\square$

## C.3. Acceleration via Meta-Bayesian Optimization

While Theorem C.4 guarantees *eventual* success, the Cyber Creator accelerates it by optimizing the search policy $\pi$. We analyze this through **No-Regret Learning**.

**Lemma C.5** (Invalid-Subspace Concentration)**.** *Suppose each reflective update prunes at least a fixed fraction $\rho \in (0, 1]$ of the currently invalid mass within $\Omega_{R_t}$. Then after $\lfloor t/K \rfloor$ reflections,*

$$\frac{|\Omega_{R_t} \cap \mathbb{G}_{bad}|}{|\Omega_{R_t}|} \ \le \ (1 - \rho)^{\lfloor t/K \rfloor} \ \xrightarrow{t \to \infty} \ 0, \tag{32}$$

*where $\mathbb{G}_{bad} = \{G \mid J(G) < \tau\}$. Thus the effective search dimension contracts geometrically with the number of meta-updates.*

*Proof.* Let $b_r = |\Omega_R \cap \mathbb{G}_{bad}|/|\Omega_R|$ after $r$ reflections. The pruning operator $\mathcal{U}_R$ removes a $\rho$-fraction of the invalid mass while preserving high-fitness regions, so $b_{r+1} \le (1 - \rho)\, b_r$. Unrolling from $b_0 \le 1$ gives $b_r \le (1 - \rho)^r$ with $r = \lfloor t/K \rfloor$, which yields (32). $\square$

**Theorem C.6** (Regret Bound Improvement)**.** *Let $R_T = \sum_{t=1}^{T} \left( J^* - J(G_t) \right)$ be the cumulative regret. With the Cyber Creator's meta-policy adapting via Multiplicative Weights Update (MWU) over $|\mathcal{A}|$ operators, the regret improves from a $\sqrt{D}$ dependence on the full search dimension $D \approx |V|^2$ to a $\sqrt{d_{eff}}$ dependence on the effective dimension $d_{eff} \ll D$:*

$$R_T^{\text{EvoMAS}} = O\big(\sqrt{d_{eff}\, T \log T}\big). \tag{33}$$

*Proof.* We decompose the bi-level regret into a (clean) meta term and a (noisy) inner term.

**1. Baseline.** A standard EA searches the full space, behaving as a $D$-armed bandit with $D \approx |V|^2$. Its minimax regret is

$$R_T^{base} = \Theta\big(\sqrt{D\, T}\big). \tag{34}$$

**2. Meta term (operator selection).** Treating the $|\mathcal{A}|$ operators as experts and running Hedge/MWU with the optimal step size $\eta = \sqrt{8 \ln |\mathcal{A}|/T}$ gives the standard guarantee (Arora et al., 2012)

$$R_T^{meta} \le \sqrt{\tfrac{T}{2}\, \ln |\mathcal{A}|} \ = O\big(\sqrt{T \ln |\mathcal{A}|}\big). \tag{35}$$

**3. Inner term (constrained search).** By Lemma C.5, the inner search is confined to the $d_{eff}$-dimensional valid manifold $\Omega_{R_t}$. Modeling it as a noisy linear search over $d_{eff}$ directions, the confidence/anytime terms contribute a logarithmic factor, so

$$R_T^{inner} = O\big(\sqrt{d_{eff}\, T\, \log T}\big). \tag{36}$$

**4. Aggregation.** Summing the two contributions and using $d_{eff} \gg \ln |\mathcal{A}|$ (so the meta term is dominated),

$$\begin{aligned} R_T^{\text{EvoMAS}} &= R_T^{meta} + R_T^{inner} \\ &\le \sqrt{\tfrac{T}{2} \ln |\mathcal{A}|} + O\big(\sqrt{d_{eff}\, T \log T}\big) \\ &= O\big(\sqrt{d_{eff}\, T \log T}\big). \end{aligned} \tag{37}$$

Comparing with the baseline (34) gives the speedup ratio

$$\frac{R_T^{\text{EvoMAS}}}{R_T^{base}} = O\left( \sqrt{\frac{d_{eff}\, \log T}{D}} \right) \xrightarrow{d_{eff} \ll D} 0, \tag{38}$$

so $R_T^{\text{EvoMAS}}$ grows strictly slower than $R_T^{base}$, proving the acceleration. $\square$

## C.4. Summary of Theoretical Claims

Collectively, our analysis establishes three advantages of EvoMAS, bridging heuristic search and rigorous optimization.

First, regarding *computational efficiency*, Proposition C.1 shows that structured operators constrain each step to a tractable local manifold, reducing the branching factor from the exponential $O(2^{N^2})$ of naive enumeration to the polynomial $O(N^2)$ of (23). This keeps every evolutionary step feasible as the agent network scales.

Second, in terms of *algorithmic reliability*, Theorem C.4 guarantees—via uniform reachability and elitism—that the system escapes local optima and reaches the $\epsilon$-optimal set almost surely even under $\delta$-bounded noise (provided $\epsilon > 2\delta$), distinguishing EvoMAS from greedy methods that may stagnate.

Finally, concerning *convergence speed*, Theorem C.6 shows that the Cyber Creator's bi-level meta-optimization, by geometrically pruning invalid regions (Lemma C.5), replaces the baseline's $\sqrt{D}$ dependence on the full dimension with a $\sqrt{d_{eff}}$ dependence on the effective dimension $d_{eff} \ll D$. The resulting bound $O(\sqrt{d_{eff} \, T \log T})$ and the vanishing ratio (38) explain why EvoMAS needs substantially fewer samples to reach high-performance workflows.

# D. Experiment Details

## D.1. Model Configuration

EvoMAS utilizes separate models for optimization and execution. For the optimizer, we employ `Claude-3.5-sonnet`, which is responsible for generating new workflows and strategies. For workflow execution and task solving, the following models are used:

- `DeepSeekV3`

- `GPT-4o-mini-0718`

- `Claude-3.5-sonnet`

- `GPT-4o-1120`

All models are accessed via official APIs. The temperature is set to 1.0 for the optimizer to encourage diverse generations and set to 0 for executors to ensure deterministic outputs. Each evolutionary run consists of 20 optimization iterations.

## D.2. Dataset Statistics

Table 12 summarizes the dataset sizes and evaluation metrics across domains.

*Table 12.* Dataset statistics and corresponding evaluation metrics.

| Domain | Dataset | #Train | #Test | Metric |
|---|---|---|---|---|
| Code Generation | HumanEval | 33 | 131 | pass@1 |
| | MBPP | 86 | 341 | pass@1 |
| Math Reasoning | GSM8K | 264 | 1055 | Accuracy |
| | MATH | 119 | 486 | Accuracy |
| Tool Used | GAIA | 94 | 372 | Accuracy |
| | ALFWorld | 27 | 107 | Success Ratio |
| Multi-hop QA | HotpotQA | 200 | 800 | F1 Score |

## E. Prompt Examples

**Example of workflow**

```python
class MultiAgentSystem:
    def __init__(self, name: str, tools=None) -> None:
        self.name = name
        self.tools = tools

    async def run(self, task: str):

        from Agent_async import Actor

        # Initialize agents
        actor = Actor(self.tools)

        # Run agent flow
        prompt = f"As a mathematician,
        solve the following complex math problem: {task}"
        res = await actor.process(prompt)

        return {"answer": res}
```

**X1: Diversity Exploration**

```python
prompt_content = f"""Create a novel multi-agent workflow for solving complex multi-
    step
reasoning problems using LLMs.

I have {num_indiv} existing agent flows with their codes as follows:
{prompt_indiv}

Please help me create a new agent flow that has a totally different form from the
    given
ones in structure and prompts. Focus on generating a workflow with completely novel
topology and agent interaction patterns, diverging from all structural templates
of the existing flows.

{self.format_prompt}
{self.agent_blocks}
{self.prompt_law}

First, describe your new agent flow and main steps in one sentence.
Then, please return the implementation of the MultiAgentSystem class in JSON format.

{{'plan': str, 'code': str, 'num_agent': int}}
"""
```

**X2: Conceptual Exploration**

```python
prompt_content = (
    f"Design a new multi-agent system for problem solving using LLMs by reimagining
    core functions from existing flows."
    f"I have {num_indiv} existing agent flows with their codes as follows:"
    f"{prompt_indiv}"
    "Please help me create a new agent flow that keeps the high-level intent or task
    decomposition ideas of the old ones, but introduces different agent types,
```

```
8      roles, and data flow structures."
9      f"{self.format_prompt}"
10     f"{self.agent_blocks}"
11     f"{self.prompt_law}"
12     "First, describe your new agent flow and main steps in one sentence. "
13     "Then, please return the Python implementation of the MultiAgentSystem class
14     in JSON format. "
15     "{'plan': str, 'code': str, 'num_agent': int}"
16 )
```

## Y1: Fine Optimization

```
1 prompt_content = (
2      f"Improve an existing high-performance agent workflow through detailed
       refinements
3      for efficiency and clarity."
4      f"I have {num_indiv} existing agent flows with their codes as follows:"
5      f"{prompt_indiv}"
6      "Please generate a new agent flow that improves on the most effective existing
7      one by optimizing prompts, reducing redundant links, and increasing information
8      clarity, while preserving the overall structure."
9      f"{self.format_prompt}"
10     f"{self.agent_blocks}"
11     f"{self.prompt_law}"
12     "First, describe your new agent flow and main steps in one sentence. "
13     "Then, please return the Python implementation of the MultiAgentSystem class
14     in JSON format. "
15     "{'plan': str, 'code': str, 'num_agent': int}"
16 )
```

## Y3: Agent Role Specialization

```
1 prompt_content = (
2      f"Design a new multi-agent system that enhances role specialization to
3      increase efficiency and collaboration."
4      f"I have {num_indiv} existing agent flows with their codes as follows:"
5      f"{prompt_indiv}"
6      "Please help me generate a novel agent flow where each agent has clearly defined
7      specialized roles and improved collaboration patterns, optimizing division of
8      labor and communication efficiency."
9      f"{self.format_prompt}"
10     f"{self.agent_blocks}"
11     f"{self.prompt_law}"
12     "First, describe your new agent flow and main steps in one sentence. "
13     "Then, please return the Python implementation of the MultiAgentSystem class
14     in JSON format. "
15     "{'plan': str, 'code': str, 'num_agent': int}"
16 )
```

## LLM-based Selection

```
1 prompt_content = (
2      f"Evaluate the following candidate agent workflows and select the optimal one "
3      f"based on structural robustness and potential for high performance."
4      f"Here are {num_candidates} candidate workflows with their code and descriptions
       :"
```

```
5      f"{candidate_descriptions}"
6      "Analyze the strengths and weaknesses of each candidate. "
7      "Select the single best workflow that balances complexity with efficiency. "
8      "Return the selected index and a brief reason for your choice in JSON format. "
9      "{'selected_index': int, 'reasoning': str}"
10 )
```

## Cyber Creator: Rule Set Meta-Reflection

```
1 prompt_content = (
2      "You are the Cyber Creator, the meta-controller of the evolutionary process. "
3      "Your goal is to refine the evolutionary search space by updating the Rule Set."
4      f"Current Rule Set: {current_rules}"
5      f"Evolution History (Last K generations): {history_log}"
6      "\n"
7      "Perform the following tasks:\n"
8      "1. [PRUNE]: Identify rules that are obsolete, conflicting, or overly
       restrictive "
9      "   based on recent high-performing workflows.\n"
10     "2. [INDUCE]: Synthesize NEW rules by abstracting common patterns found in "
11     "   the top 10% of workflows.\n"
12     "\n"
13     "Return the updates in JSON format containing 'rules_to_remove' (list of IDs) "
14     "and 'rules_to_add' (list of triplets: condition, weight, description)."
15 )
```

## Cyber Creator: Strategy Distribution Adaptation

```
1 prompt_content = (
2      "Analyze the efficacy of evolutionary operators in the recent epoch."
3      f"Operator Statistics: {operator_stats}"
4      "(Format: {operator_name: avg_fitness_gain, success_rate, cost_efficiency})"
5      "\n"
6      "Based on the Multiplicative Weights Update (MWU) principle, suggest adjustments
        "
7      "to the strategy probability distribution.\n"
8      "- Amplify strategies that yield high fitness gains with low cost.\n"
9      "- Dampen strategies that lead to stagnation or errors.\n"
10     "\n"
11     "Provide a reasoning for the shift and the new target weights (must sum to 1.0).
        "
12     "Output JSON: {'reasoning': str, 'new_weights': {op_name: float}}"
13 )
```

