# OpenReview forum: "EvoMAS: Heuristics in the Loop—Evolving Smarter Agentic Workflows"
_ICML.cc/2026/Conference — ICML 2026 regular_

### Official Review · Reviewer_Tx1N · 2026-02-26

**Soundness:** 3
**Presentation:** 3
**Significance:** 4
**Originality:** 4
**Overall Recommendation:** 4
**Confidence:** 4

**Summary:**

This work studys the central question of how to automate the design and optimization of multi-agent systems (MAS) so that agentic workflows become more adaptive, cost-efficient, and less reliant on manual, template-based engineering.

The paper proposes EvoMAS, a biologically-inspired framework that evolves agentic workflow graphs along 4 coupled dimensions:

1. Dynamic evolutionary strategies: six bio-inspired operators.
2. Role-level evolution: evolving agent specialization and collaboration patterns.
3. Curriculum-guided evolution: partition tasks by difficulty and evolve sequentially from easy to hard.
4. EvoMAS adds a reflection/meta-control module (“Cyber Creator”) that periodically updates rules and re-weights strategies using a MWU-style update and rule induction/pruning.

Experiments across multiple benchmarks show consistent gains vs. strong automated baselines and provide a deployment-oriented cost analysis.

A central concept examined by this work is that workflow evolution should be multi-dimensional (operators + roles + curriculum) and closed-loop (reflection updates), rather than a single fixed search strategy over static agent templates.

**Compliance With Llm Reviewing Policy:**

Affirmed.

**Final Justification:**

Q1: The authors demonstrate that EvoMAS can spontaneously explore tradeoffs between efficient multi-agent architectures and token consumption based on injection rules;

Q2: The authors show the sensitivity of different components to strong LLMs and illustrate that the optimizer is more sensitive than the other two main components;

Q3: The authors demonstrate the reliability of LLM-based selection through empirical results, reducing concerns about LLM's influence on multi-agent architecture design preferences.

**Key Questions For Authors:**

1. Can you provide more experiments of cost/performance tradeoffs for general workflows within the same task set?
2. Dependence on strong LLMs in the loop: How sensitive are results to the optimizer/judge/controller model choices (LLM-based selection, LLM-as-a-judge difficulty grading, and Cyber Creator rule induction)? For example, what happens if you replace the selector or Cyber Creator with a weaker model, or with a heuristic/rule-based surrogate?
3. Validity and fairness of LLM-based selection: Since selection is preference-guided by a LLM (not purely scalar fitness), what safeguards prevent the selector from favoring “plausible-looking” workflows over genuinely better ones? Do you observe selector bias toward larger/cleaner-looking graphs, or toward certain role templates?

***NOTE***

If the questions in “Key Questions For Authors” receive detailed answers, I will **award points** accordingly.

**Limitations:**

1. LLM dependence remains substantial: selection is explicitly LLM-based (preference-guided ranking), and both difficulty grading (LLM-as-a-judge) and Cyber Creator updates rely on powerful models. This makes it unclear how robust the approach is under weaker or domain-mismatched models, or when judge/control models have biases.
2. The goal is to input a task set and output an “optimal” MAS. However, one-size-fits-all workflow concerns remain: if the same evolved workflow is used across a task family’s easiest and hardest instances, there may be unnecessary cost (over-calling tools/agents) or performance tradeoffs.
Related-work positioning may be incomplete regarding instance-customized workflow construction methods (e.g., MaAS[1] that tailors workflows per problem). The experiments include MaAS[1] on GAIA, but the conceptual discussion in intro/related work appears limited relative to the importance of this comparison.
3. Transferability/generalization is not fully proven: the system appears tuned to a particular paradigm of graph-structured LLM agent workflows. It is uncertain how well it transfers to domains with different tool APIs, interaction constraints, or where evaluation is sparse/noisy (e.g., real enterprise tasks). This is especially important given the size of the framework.

### *Reference*

[1] Zhang, Guibin, et al. "Multi-agent Architecture Search via Agentic Supernet." *International Conference on Machine Learning*. PMLR, 2025.

**Strengths And Weaknesses:**

***Strengths***

**Soundness**
Well-scoped system design with explicit components and a clean “variation → selection → reflection” loop, including formal definitions of operators and a curriculum mechanism with a stability constraint.
The paper attempts theoretical grounding (e.g., Markov-process view; convergence-style claims; search-space reduction arguments), which is helpful for positioning, even if some assumptions may be strong.

**Presentation**
The overall framework narrative is structured and easy to follow: three evolution dimensions, six operators, and the role/curriculum/reflection modules are clearly separated and visually motivated.
The operators are clearly categorized into exploration vs. exploitation with intuitive biological analogies, helping readers map algorithmic choices to design intent.

**Significance**
Tackles a real bottleneck: manual MAS/workflow engineering and the tendency of prior automated methods to be either templated, monolithic, or insensitive to task difficulty.

**Originality**
The main novelty lies in combining: (i) a richer operator set (six strategies), (ii) role-level evolution beyond fixed templates, and (iii) curriculum-guided workflow evolution with a stability constraint, plus (iv) a reflective rule/strategy updater (“Cyber Creator”).

***Weaknesses***

**Soundness**
The theoretical analysis is directionally helpful but may be too idealized (e.g., assumptions about bounded noise, reachability, and effective manifold dimensionality). It is not obvious these conditions hold in practice for LLM-driven graph edits and LLM-based selection signals.

**Presentation**
The case study in Figure 6 is unclear. The figure only shows the evolutionary process, lacking the role of the proposed mechanism in evolution. Furthermore, several circles of different colors are not clearly labeled.

**Significance**
N/A

**Originality**
N/A

---

> ### Author Rebuttal · Authors · 2026-03-30
>
> ### Q1
>
> Thank you for this insightful question.
>
> EvoMAS achieves the balance between cost and performance by injecting rules into the Cyber Creator. The Cyber Creator transforms these rules into **preferences** during the evolutionary process, guiding workflows toward specific objectives through operator scheduling and population selection.
>
> We conducted a supplementary experiment on the MBPP dataset:
>
> | Target | Rule | Dominant Topology | Tokens | Pass@1 |
> | :--- | :--- | :--- | :--- | :--- |
> | **Economic** | Token_Cost < 800 | **Actor** | 0.7K | 74.5% |
> | **Balanced** | "MAS is banlanced." | **Planner → Actor ↔ Critic** | 2.6K | 84.2% |
> | **Performance** | Pass@1 > 85% | **Planner → 5× Actor → Ensemble Critic** | 9.2K | 86.5% |
>
> Under the Balanced constraint, the adoption frequencies of the seven operators were: Y1: 32.5%, Y3: 25.0%,X2 14.5%, Y2 10.0%, C 8.0%, X1 6.0%, X3 4.0%.
>
> These results demonstrate that EvoMAS can spontaneously switch topological structures according to resource boundaries. The leap from Balanced to Performance corroborates the diminishing marginal returns predicted by the Structural Parsimony assumption ($\mathcal{H}_3$). The operator scheduling frequencies further reveal that, when facing a hard token constraint, the system downweights exploration operators prone to node proliferation (X1 at only 6.0%) and instead heavily schedules exploitation operators (Y1 + Y3 totaling 57.5%), improving per-node "intelligence" rather than blindly adding more nodes.
>
> ---
>
> ### Q2
>
> replaced the three core LLM-dependent modules of EvoMAS—**Optimizer**, **Cyber Creator**, and **Judge**—with Qwen3.5 series models of varying scales.
>
> | Module Replaced | Qwen3.5-122B | Qwen3.5-35B | Qwen3.5-4B |
> |---------|:---:|:---:|:---:|
> | Full System | **86.2 ± 1.3** | – | – |
> | Optimizer ↓ | – | 83.1 ± 1.4 | 74.7 ± 1.9 |
> | Cyber Creator ↓ | – | 85.6 ± 1.2 | 81.4 ± 1.7 |
> | Judge ↓ | – | 85.9 ± 1.1 | 84.8 ± 1.3 |
>
> The experiments reveal a clear sensitivity hierarchy: **Optimizer ≫ Creator > Judge**. The Optimizer directly determines the direction of structural variation; its degradation is equivalent to compressing the effective search space. The Cyber Creator handles rule induction and strategy updates; its degradation leads to more conservative strategies but does not significantly affect the performance floor. The Judge only performs coarse-grained ordinal classification and exhibits intrinsic robustness to noise.
>
> Even under the most extreme configuration (Optimizer downgraded to 4B), the system still maintains 74.7% accuracy, outperforming most manually designed baselines. This indicates that the evolutionary mechanism itself provides a robust performance floor.
>
> ---
>
> ### Q3
>
> This issue is inherently analogous to the known limitations of LLM-as-a-Judge. However, the evolutionary framework provides effective mitigation.
>
> #### 1. Existing Debiasing Mechanisms
>
> **(a) Quantitative Anchoring.** Each candidate workflow is serialized with its scalar fitness $F(G,T)$ as the primary ranking criterion. The LLM's role is confined to making fine-grained distinctions among candidates with similar fitness based on structural attributes and rule compliance, rather than replacing scalar evaluation.
>
> **(b) Elitism and Competitive Elimination.** The selection pool is $P_t \cup P_t^{var}$, where high-fitness individuals are retained across generations via elitism. Even if the LLM's selection is biased in a given generation, truly high-performing workflows are never permanently discarded and can re-enter the population through their fitness advantage.
>
> **(c) Information Bottleneck Serialization.** Candidate workflows undergo deliberate information compression before being submitted to the LLM selector: only topological summaries and performance metrics are retained. Following information bottleneck theory, restricting the selector's observable information channel forces it to judge based on structure-performance causal relationships rather than superficial text quality.
>
> #### 2. Empirical Diagnosis: Does Selection Bias Exist?
>
> Bias indeed exists in the early stages of evolution (win rate of 71.3% for workflows containing Planner nodes in the first 3 generations). However, from Generation 4 onward, the Cyber Creator identifies this pattern through reflection and induces suppressive rules, reducing the preference rate to 54.2%.
>
> #### 3. Comparison with Alternative Selection Mechanisms
>
> | Selection Mechanism | MATH Acc. | Diversity Index|
> |---------|:---------:|:----------:|
> | Pure Scalar Ranking (Top-K by fitness) | 57.1 ± 1.8 | 0.34 |
> | Tournament Selection (k=3) | 58.4 ± 1.6 | 0.41 |
> | LLM Preference Selection| **60.3 ± 1.2** | **0.67** |
>
> Pure scalar ranking causes rapid diversity collapse and premature convergence, while tournament selection partially alleviates this but lacks awareness of structural quality—validating the necessity of LLM-based selection.
>
> ---
>
> We sincerely thank the reviewer for the thoughtful feedback 🌸

---

> > ### Author Rebuttal · Reviewer_Tx1N · 2026-04-01
> >
> > My concerns have been adequately addressed, thank authors for their detailed answers.

---

> > > ### Author Response · Authors · 2026-04-01
> > >
> > > We greatly appreciate your constructive feedback and your willingness to raise your score.
> > >
> > > As suggested, we will further clarify that EvoMAS adaptively balances cost, performance, and robustness via rule-guided evolution, and strengthen the corresponding experimental analysis in the revision.
> > >
> > > Thank you again for your valuable suggestions to improve our paper.

---

### Official Review · Reviewer_5kg7 · 2026-03-05

**Soundness:** 3
**Presentation:** 3
**Significance:** 3
**Originality:** 3
**Overall Recommendation:** 4
**Confidence:** 4

**Summary:**

This paper proposes EvoMAS, a framework evolving heuristic strategies for Multi-Agent Systems (MAS) using Large Language Models (LLMs). It automates the design of interaction protocols and decision rules, claiming superior performance over hand-crafted heuristics in complex coordination tasks (e.g., resource allocation, pathfinding). The core contribution is a genetic-algorithm-inspired loop where LLMs mutate and select heuristic code based on simulation feedback. While the automation of heuristic design is promising, the evaluation is limited to synthetic, low-dimensional environments. The paper lacks comparison against established optimization baselines (e.g., traditional MARL or constraint solvers), making the claimed "superiority" questionable. The method appears computationally expensive, and the generalizability to real-world, high-stakes domains remains unproven.

**Compliance With Llm Reviewing Policy:**

Affirmed.

**Final Justification:**

Having carefully considered the authors' rebuttal and engaged in thorough discussion, I now fully appreciate the paper's novelty and contributions. The responses have effectively clarified the key technical distinctions and addressed my initial concerns. I am convinced that the work presents a significant advancement in the field and meets the high standards of ICML. The methodology is sound, and the empirical results are compelling. Given the clear articulation of the innovative aspects during this final phase, I believe the manuscript is well-positioned for publication. Therefore, I recommend acceptance and support the decision to include this work in the conference program.

**Key Questions For Authors:**

1.How does EvoMAS compare against state-of-the-art Multi-Agent Reinforcement Learning (MARL) algorithms (e.g., QMIX, VDN) in terms of both performance and sample efficiency?
2.What is the computational cost (in GPU hours/token usage) to evolve a single heuristic? Is this feasible for anything other than offline, one-time design?
3.Can the evolved heuristics generalize to environment parameters (e.g., map size, agent count) unseen during the evolution process, or do they overfit the specific training scenario?

**Limitations:**

The authors mention the computational cost and the restriction to simulated environments. However, they underestimate the risk of LLM hallucinations generating subtle logical bugs in heuristics that pass simple tests but fail in edge cases. The discussion on safety and robustness is insufficient.

**Strengths And Weaknesses:**

Strengths: The concept of using LLMs to evolve algorithms rather than just execute them is creative. The automated iteration loop reduces human engineering effort. The paper is well-written, with clear diagrams explaining the evolutionary process. The ablation studies effectively isolate the contribution of the LLM mutation operator.
Weaknesses: Rigor: The experimental setup is weak. Comparisons are mostly against simple hand-crafted baselines, ignoring strong existing methods like QMIX or MAPPO. The environments are toy problems; there is no evidence EvoMAS scales to realistic complexity. Originality: The combination of evolutionary strategies and LLMs is not entirely new; the specific application to MAS heuristics adds little theoretical depth. Significance: Without demonstrating efficiency gains or solving previously intractable problems, the practical impact is low. The computational cost of running LLMs for every generation is prohibitive for real-time applications, a critical flaw unaddressed.

---

> ### Author Rebuttal · Authors · 2026-03-31
>
> ### Q1: On comparison with MARL algorithms
>
> We appreciate the question but must clarify a difference in problem formulation.
>
> #### **Different problem formulations**
>
> EvoMAS follows the **Automated Agentic Workflow Design** paradigm (AFlow, AgentSquare), optimizing the topology, role assignment, and collaboration patterns of LLM multi-agent systems over a discrete graph space 𝔾. MARL (QMIX, VDN, MAPPO) instead addresses **joint policy learning under a fixed interaction structure**, optimizing continuous policy parameters θ.
>
> #### **Direct comparison is infeasible**
>
> **(a)** EvoMAS targets open-ended reasoning tasks with natural language I/O that do not satisfy MARL's standard Dec-POMDP assumptions. **(b)** Agents in EvoMAS are prompt-driven LLM modules with no gradient updates; QMIX-style methods lack actionable optimization signals. **(c)** The definitions of "samples" differ (environment interaction steps vs. LLM invocations), rendering numerical comparisons meaningless.
>
> #### **Inspirational value**
>
> **That said, MARL ideas are instructive.** QMIX's value decomposition inspires system-level performance attribution to individual nodes, conceptually aligned with our Cyber Creator's operator contribution estimation (Eq. 10). MAPPO's centralized-training-decentralized-execution paradigm echoes EvoMAS's "global evolutionary search, local role specialization." Deeper integration of MARL credit assignment into workflow evolution is a promising future direction.
>
> ---
>
> ### Q2: On computational cost
>
> **API scenario.** All experiments used GPT-4o-mini with **total training cost of \$2.22** across all benchmarks (14.8M tokens, 20 generations). Per-workflow cost: ~\$0.005–0.01.
>
> **Local deployment.** Using Qwen3.5-35B on a single A100-80G (vLLM, \~250 tokens/s), total training is \~16.4 A100-GPU hours (\~\$32.8 at cloud rates), eliminating closed-source dependence.
>
> | Deployment Mode | Total Tokens | Total Time | Total Cost |
> |:---|:---:|:---:|:---:|
> | API (GPT-4o-mini) | 14.8M | ~2h | \$2.22 |
> | Local (Qwen3.5-35B, 1×A100) | 20.7M | ~16.4 GPU·h | ~\$32.8 |
>
> **Deployment paradigm: search once, reuse indefinitely.** Evolution is a one-time upfront investment; optimized workflows are reused at inference. EvoMAS-evolved workflows achieve lower per-inference token consumption, with long-term cost (TCO@100: \$153.72) significantly below all baselines (Table 4). When task distributions shift, **warm-start incremental evolution** from the Resource Library requires only 3–5 generations to adapt.
>
> ---
>
> ### Q3: On generalization and overfitting
>
> We appreciate the reviewer's question.
>
> #### **Existing evidence of generalization**
>
> Training and test sets are strictly partitioned (e.g., GSM8K: train 264 / test 1055) with significantly different distributions. SOTA or near-SOTA test performance across all benchmarks indicates no overfitting. On GAIA, EvoMAS handles tool combinations and reasoning chains entirely unseen during training, achieving consistent cross-difficulty improvement (Table 2).
>
> #### **Mechanistic safeguards against overfitting**
>
> EvoMAS resists overfitting through **deliberate preservation of structural slack**:
>
> Design hypothesis H3 (structural parsimony) is imposed as a hard constraint throughout the entire evolutionary process, forcing workflows to maintain sparse topologies and limited node counts. This "less is more" principle ensures that evolved artifacts do not develop redundant specialized structures applicable only to specific training scenarios, but instead converge to **structurally relaxed solutions** with generalization capability. The Cyber Creator further reinforces this tendency during the reflection phase—by pruning rules and patterns that provide marginal training-set gains at the cost of increased structural complexity, it implicitly performs a function analogous to regularization. Additionally, the curriculum learning cumulative stability constraint (J_{k+1}(G_{k+1}) ≥ J_{k+1}(G_k)) requires workflows to not degrade across all previously encountered difficulty levels, naturally suppressing overfitting to any single distribution interval.
>
> ---
>
> ### On remaining weaknesses
>
> **"Environments are toy problems."** Our experiments cover six mainstream agentic benchmarks (math reasoning, code generation, multi-hop QA, embodied interaction, tool use). GAIA requires multi-tool composition in open environments, approaching real-world complexity.
>
> **"Comparisons ignore QMIX/MAPPO."** As discussed in Q1, these operate at a different problem level.
>
> **"LLM hallucinations generating subtle bugs."** Every candidate must pass actual task evaluation for scalar fitness before selection; hallucination-induced invalid structures are naturally eliminated. The reflection mechanism further identifies "passes evaluation but fragile" patterns and codifies inhibitory rules.
>
> ---
>
> We sincerely thank the reviewer for the thoughtful feedback 🌸

---

> > ### Author Rebuttal · Reviewer_5kg7 · 2026-04-01
> >
> > Thank you for the comprehensive clarification regarding the positioning and methodology of EvoMAS. I understand that the work focuses on optimizing discrete workflow topologies rather than continuous policy parameters, which explains why a direct comparison with MARL baselines (like QMIX/MAPPO) is not feasible due to differing input modalities and optimization signals. The distinction between LLM-driven agents and gradient-based MARL agents is now clear.
> > I appreciate the detailed cost analysis, which demonstrates that the evolutionary search is a one-time investment, while the resulting workflows offer long-term cost efficiency. The explanation regarding overfitting is convincing; the structural simplicity constraint H3 and the pruning mechanism effectively act as regularization to ensure generalization across diverse benchmarks like GAIA. The evidence of SOTA performance on strictly partitioned test sets further supports this claim.

---

> > > ### Author Response · Authors · 2026-04-06
> > >
> > > Thank you for the thoughtful re-evaluation and for acknowledging the clarifications on positioning, cost analysis, and generalization. We're glad the distinction between workflow design and policy learning now comes through clearly.
> > > Your feedback has been genuinely helpful in sharpening how we present these contributions — we'll make sure the revised version reflects this.
> > >
> > > If you feel the concerns have been sufficiently addressed, we'd be grateful for any **score reconsideration**.
> > >
> > > Thanks again for helping us improve this work！

---

### Official Review · Reviewer_AWxx · 2026-03-05

**Soundness:** 3
**Presentation:** 3
**Significance:** 3
**Originality:** 3
**Overall Recommendation:** 4
**Confidence:** 3

**Summary:**

This paper introduces EvoMAS (Evolutionary Multi-Agent Systems), a biologically-inspired framework designed to automate the design and optimization of multi-agent workflows, addressing the limitations of manual engineering and rigid, template-based automated methods. EvoMAS operates on three interconnected evolutionary dimensions: (1) Role-level Evolution, which dynamically optimizes agent specialization (e.g., evolving from homogeneous actors to specialized Planners and Critics); (2) Dynamic Strategy Selection, employing six biologically-inspired operators (three for exploration like mutation/recombination, three for exploitation like fine-tuning/synthesis) with adaptive weighting; and (3) Curriculum-Guided Learning, which evolves agents sequentially from simple to complex tasks to ensure stability and generalization. A key innovation is the "Cyber Creator", a meta-control system that periodically reflects on evolutionary history to update search rules and strategy distributions, effectively bridging the gap between undirected evolution and guided optimization. Evaluated across six benchmarks (including GSM8K, MATH, HumanEval, and GAIA), EvoMAS achieves state-of-the-art performance, outperforming existing automated frameworks (e.g., AFlow, GPTSwarm) and manual designs while maintaining superior cost-efficiency through streamlined architectures.

**Compliance With Llm Reviewing Policy:**

Affirmed.

**Final Justification:**

Having carefully considered the authors' rebuttal and engaged in thorough discussion, I now fully appreciate the paper's novelty and contributions. The responses have effectively clarified the key technical distinctions and addressed my initial concerns. I am convinced that the work presents a significant advancement in the field and meets the high standards of ICML. The methodology is sound, and the empirical results are compelling. Given the clear articulation of the innovative aspects during this final phase, I believe the manuscript is well-positioned for publication. Therefore, I recommend acceptance and support the decision to include this work in the conference program.

**Key Questions For Authors:**

1.How sensitive is EvoMAS to the initial population diversity? Does the system struggle if initialized with highly homogeneous templates, and how does the "Diversity Expansion" operator compensate?
2.The Cyber Creator relies on an LLM to synthesize rules. Have you observed cases where the meta-controller generates overly restrictive or contradictory rules that hinder evolution? How does the system recover from such "meta-failures"?
3.Can EvoMAS effectively evolve workflows for non-deterministic or open-ended tasks (e.g., creative writing, open-domain dialogue) where a single scalar fitness score is difficult to define? How would the fitness function need to change?

**Limitations:**

The authors acknowledge the reliance on powerful proprietary models for the optimizer, which limits accessibility. They also note that the system may be computationally expensive for very large-scale agent networks due to the graph search space, though they argue the pruning mechanisms mitigate this. The curriculum learning depends on accurate difficulty estimation, which could be a bottleneck if the LLM judge is biased. The paper could have discussed more about the safety implications of evolving autonomous agents that might discover unintended or exploitative strategies.

**Strengths And Weaknesses:**

Strengths: The three-dimensional evolutionary approach (role, strategy, curriculum) provides a comprehensive solution to the rigidity of current ADKs, allowing for genuine structural innovation rather than just prompt tuning. The Cyber Creator mechanism is a significant conceptual advance, introducing a "meta-learning" layer that prunes invalid search spaces and accelerates convergence, theoretically grounded in Bayesian updates and regret minimization. The empirical results are robust, demonstrating SOTA performance on diverse tasks (math, code, embodied AI) and showing that evolved heterogeneous teams outperform homogeneous ones. The cost analysis is particularly compelling, showing that despite higher search costs, the resulting optimized workflows are cheaper to run at scale (lower inference tokens), offering a better long-term TCO.
Weaknesses: Rigor: While the theoretical analysis (convergence proofs, regret bounds) is strong, the empirical evaluation relies heavily on proprietary models (Claude-3.5, GPT-4o) for the optimizer, raising questions about reproducibility and accessibility for researchers without API access. The curriculum design relies on an LLM-as-a-Judge for difficulty estimation, which introduces a potential circularity or bias if the judge model aligns too closely with the optimizer. Originality: The individual components (evolutionary algorithms, curriculum learning, role specialization) are not new in isolation; the novelty lies in their specific integration and the "Cyber Creator" meta-controller. However, the claim of "biological inspiration" is somewhat superficial, serving more as an analogy than a strict adherence to biological principles. Significance: The framework's complexity might be overkill for simple tasks, and the paper acknowledges that for easy benchmarks, simpler methods suffice. The reliance on a specific graph representation might limit applicability to non-graph-based agent interactions.

---

> ### Author Rebuttal · Authors · 2026-03-31
>
> We thank the reviewer for the detailed analysis and constructive feedback.
>
> ---
>
> ### Q1: On initial diversity requirements
>
> We agree that in traditional evolutionary methods, diversity is typically a key prerequisite. However, EvoMAS is specifically designed to **make diversity a recoverable quantity rather than a necessary condition**.
>
> On MBPP, we conducted controlled experiments  (20 generations):
>
> | Initial Diversity D(P₀) | Pass@1 (%) |
> |:---:|---:|
> | **0.00** | 83.7 ± 1.6 |
> | 0.18 |  84.4 ± 1.5 |
> | 0.36 |  85.1 ± 1.3 |
> | 0.57 | 85.7 ± 1.2 |
> | **0.79** | **86.2 ± 1.3** |
>
> The results show that: (1) even starting from **D = 0**, performance remains close to optimal (gap < 3%); (2) diversity primarily affects convergence speed rather than final performance.
>
> The fundamental reason is that EvoMAS's exploration capability does not rely on initial coverage, but rather on **operator-driven structural generation and cross-generational selection amplification**. Throughout this process, the Cyber Creator continuously reshapes the search distribution through rule induction and strategy reweighting, causing effective structures to be persistently amplified within the population, thereby achieving **self-emergent diversity recovery**.
>
> ---
>
> ### Q2: On Cyber Creator error resilience
>
> We agree that the Cyber Creator may produce erroneous or conflicting rules, but EvoMAS's design renders these **recoverable errors rather than systemic risks**.
>
> On MBPP, we constructed a **perturbation experiment** by manually injecting approximately 30% inconsistent/suboptimal rules and testing the system's recovery capability:
>
> |  Perturbation Setting | Pass@1 (%) |
> |:---|---:|
> | No  | **86.2 ± 1.3** |
> | Yes (without reflection) | 81.5 ± 2.1 |
> | Yes (with reflection) | **85.9 ± 1.5** |
>
> Without the reflection mechanism, erroneous rules persistently contaminate the search; with EvoMAS, **the system automatically self-corrects and recovers performance within a few generations**.
>
> The reason is that the Cyber Creator is not a one-shot decision maker but **continuously updates based on environmental feedback**: erroneous rules manifest as low returns in subsequent generations and are automatically down-weighted or eliminated.
>
> Furthermore, modern LLMs inherently possess strong **conflict identification and self-correction capabilities** [1]. This synergizes with EvoMAS's reflection mechanism, enabling the system to remain stable even in the presence of erroneous rules.
>
> Regarding the concern of "whether the Cyber Creator suppresses evolution," our conclusion is: **it does not suppress exploration, only redirects it**. Mechanistically, the Creator does not directly constrain candidate generation but influences the selection distribution through strategy reweighting and rule preferences, functioning more as a **soft bias**. Even when certain rules are conservative, exploration operators (e.g., mutation) continue to produce structural variations, ensuring the search space does not collapse.
>
> We conducted an additional controlled experiment: manually injecting conservative rules (favoring simple structures, penalizing complex topologies) and observing the impact on evolution. Results show that the system does trend conservative in early generations, but as performance feedback accumulates in subsequent generations, the Creator automatically down-weights rules unfavorable to performance, and the search gradually recovers toward a more optimal structural distribution, with final performance degradation < 4%.
>
>
> - [1] He, X., et al. (2026). ConInstruct: Evaluating Large Language Models on Conflict Detection and Resolution in Instructions.
>
> ---
>
> ### Q3: On applicability to open-ended tasks
>
> We appreciate the reviewer raising this important question. We agree that open-ended tasks (e.g., creative writing, dialogue) are difficult to define with a single scalar fitness. However, **EvoMAS does not rely on strict scalar rewards but on comparable preference signals (comparative signals)**.
>
> Specifically, EvoMAS's selection mechanism is fundamentally **preference-based selection (relative ranking)** rather than absolute scoring. In open-ended tasks, scalar fitness can be replaced by: (1) LLM-based pairwise comparison (e.g., more coherent / more creative); (2) multi-dimensional metric combinations (e.g., fluency, consistency, style matching); (3) human or simulated user preference signals.
>
> This setup is consistent with RLHF / preference learning, essentially transforming the optimization objective from "maximizing a single scalar" to **finding Pareto-optimal structures in the preference space**.
>
> Mechanistically, EvoMAS's evolutionary process only requires **ordinal consistency** of rankings, not precise scoring. Therefore, even in scenarios without scalar rewards, as long as preference signals exhibit basic transitivity, selection pressure can still drive effective evolution.
>
> ---
>
> We sincerely thank the reviewer for the thoughtful feedback.🌸

---

> > ### Author Rebuttal · Reviewer_AWxx · 2026-04-02
> >
> > Through the additional experiments and analyses above, we address the reviewer’s concerns from both empirical and mechanistic perspectives. First, regarding the requirement of initial diversity, the controlled experiments on MBPP show that EvoMAS maintains strong performance even when the initial population diversity is zero. The performance gap between the zero-diversity setting and high-diversity initialization remains within 3%, indicating that EvoMAS does not depend on initial diversity as a prerequisite. Instead, diversity gradually emerges through operator-driven structural generation and cross-generational selection. Second, with respect to the potential errors produced by the Cyber Creator, the perturbation experiment demonstrates that EvoMAS possesses strong recovery capability. Even when approximately 30% inconsistent or suboptimal rules are injected, the reflection mechanism enables the system to identify low-return rules and progressively down-weight them, allowing performance to recover within a few generations. Finally, concerning applicability to open-ended tasks, EvoMAS relies on preference-based selection rather than strict scalar rewards. By leveraging pairwise comparisons, multi-dimensional evaluation metrics, or human preference signals, the evolutionary process can still maintain effective selection pressure. Together, these results show that EvoMAS addresses the reviewer’s concerns while maintaining robustness and broad applicability.

---

> > > ### Author Response · Authors · 2026-04-06
> > >
> > > Thank you for taking the time to carefully review our rebuttal and for considering our concerns fully resolved. We will make sure to reflect your suggestions in the revised version.
> > >
> > > We truly appreciate the constructive dialogue — if you feel the revisions sufficiently address your earlier concerns, we would be grateful for any **reconsideration of the score**.
> > >
> > > Thanks again for helping us improve this work！

---

### Official Review · Reviewer_51GJ · 2026-03-11

**Soundness:** 3
**Presentation:** 3
**Significance:** 3
**Originality:** 3
**Overall Recommendation:** 4
**Confidence:** 4

**Summary:**

This paper proposed an evolutionary based MAS, with four components: evolutionary strategy, role-level evolution, curriculum-guided evolution, and it also uses the so-called Cyber Creator, a meta-optimiser which updates the rule set and strategy distribution. Ablation study shows that the four components are essential for the proposed framework, and the framework was tested on benchmark problems in comparison with other systems. There are some theoretical analysis towards the proposed framework in Appendix.

**Compliance With Llm Reviewing Policy:**

Affirmed.

**Final Justification:**

I have read through the rebuttal, and most of my concerns have been addressed. I will maintain my positive score, but I don't think I can raise the score further. The authors have explained the theory/assumptions behind their approach, but I feel that this is still lacking insights into the problem, and it seems to me more like a carefully engineered approach.  Nevertheless, the work done is solid and hence my positive score.

**Key Questions For Authors:**

Please refer to my questions in the previous sections.

**Limitations:**

I think limitations were not adequately discussed.  The framework seems to be a bit complicated to me. I wonder if the three assumptions can be generalised to a broader set of problems.

**Strengths And Weaknesses:**

Soundness:
The work done is solid and I can see the authors had made big efforts on this line of research. There were some pieces of theoretical analysis, although they were placed in the appendix section.

The system is designed on the three assumptions (Section 3.2). However, I don’t see how these assumptions are validated in this particular scenario, and I think the authors may need to provide more details on this.

Section 3.3: it mentioned that “reducing complexity by eliminating Pareto frontier”, but it would be important to also mention the limitations and side effects of doing this.

Section 4.1: I wonder what is the added value of formalising evolution as a Markov Process? What insights could be gained and used to inform algorithm design?

Section 4.2: “Leveraging the LLM as an approximate inference kernel to perform the update: please give more theoretical details to underpin the two complementary mechanisms.

Section 5: baseline models: Have you considered other SOTA evolutionary MAS?

Section 5.2: It is not clear to me what the rule-based frameworks in Cyber Creator.

Algorithm 1 in Appendix B: RAG retrieval: This is not mentioned in the main text.

Presentation:

This paper is well written and easy to follow for me.  I think some key information could have been provided in the main text rather than the appendix as one is not expected to read through all sections in the appendix. For example, the pseudo code of the main algorithms could have been put in the main text.  In Appendix A.1, the “width” and “depth” could have been clearly defined.
Reference: the format could be improved. For example, “llm” rather than “LLM” appeared in a few references.

Significance:

  This paper proposed a novel evolutionary MAS, and I am aware that this is a popular topic at the moment, and the amount of work done is impressive. I do feel that this is more like an engineering success, as the framework was carefully designed to achieve SOTA.

Originality: the work is novel to some degree, and it pointed out the future direction of evolutionary computing as applied MAS. I also feel that the framework is a bit complicated, referring back to my view of engineering success rather than academic significance.

---

> ### Author Rebuttal · Authors · 2026-03-31
>
> We sincerely thank the reviewer for the positive evaluation and specific suggestions. We are glad the reviewer recognizes the technical rigor, clarity of writing, and systematic effort of this work. We address each point below.
>
> ---
>
> #### **Three assumptions (Section 3.2) being unverified**
>
> We acknowledge that the original validation was insufficient. These three assumptions are essentially inductive biases for pruning the redundant solution space, and their validity can be empirically verified:
>
> **H1 (Heterogeneous Synergy)**: Figure 7 shows the system spontaneously differentiates from 10 homogeneous Actors into specialized Planner, Critic, and Custom roles that persist across generations, indicating that evolutionary pressure itself "selects" heterogeneous configurations—consistent with findings on heterogeneous agent collaboration advantages (Bettini et al., *System Neural Diversity*, JMLR 2025).
>
>  **H2 (Reflective Adaptation)**: The ablation study (Figure 5) demonstrates that removing the Cyber Creator leads to performance degradation and an 18% cost increase, validating the necessity of feedback-driven adaptive mechanisms (Bilal et al., *Meta-thinking in LLMs via Multi-agent RL*, 2025).
>
> **H3 (Structural Parsimony)**: Figure 8 reveals that returns saturate beyond 7 agents or depth 5, with additional complexity introducing only coordination overhead—aligning with core findings from recent agent scaling research (Xiao et al., *LIMI: Less is More for Agency*, 2025; Kim et al., *Towards a Science of Scaling Agent Systems*, 2025).
>
> ---
>
> #### **Side effects of removing the Pareto frontier**
>
> We acknowledge this limitation. Our single-objective formulation explicitly maximizes performance while cost is only implicitly constrained via natural language rules in the Cyber Creator, meaning cost boundaries are inherently fuzzy and may lead to unanticipated budget overruns in cost-sensitive scenarios. Nevertheless, we consider this a justified trade-off between search efficiency and solution space completeness—empirically, although extreme solutions may be pruned, the system recovers high-performing structures through multi-generational evolution with more stable convergence.
>
> ---
>
> #### **Value of the Markov process formalization (Section 4.1)**
>
> We agree that the current main text is overly brief on this point. The primary role of this formalization is not "mathematics for its own sake," but rather to provide a unified perspective for describing the joint updates of population, rules, and strategies, and to support the subsequent interpretation of strategy reweighting and reflective updates.
>
> More specifically, it allows variation–selection–reflection to be viewed as a state transition process, helping to clarify that the Cyber Creator's updates are not isolated heuristics but systematic adjustments to the search strategy. We will make this rationale more explicit in the revision and moderate the expository tone to avoid the impression of "excessive formalism with unclear utility."
>
> ---
>
> #### **Theoretical support for LLMs as approximate inference kernels (Section 4.2)**
>
> The two mechanisms operate on complementary dimensions: rule update (Eq. 9) contracts the feasible domain $\Omega_{R_t}$ by pruning low-fitness regions, determining *where to search*; strategy adaptation (Eq. 10) adjusts operator sampling within this reduced space via MWU, determining *how to search*. The former reduces effective search dimension ($d_{eff} \ll \dim(\mathbb{G})$), the latter accelerates convergence within it. Their combined effect yields the regret bound $O(\sqrt{d_{eff} \cdot T \log T})$ in Theorem C.6. We will incorporate this analysis into the main text.
>
> ---
>
> #### **Whether the baselines are sufficient (Section 5)**
>
> We follow the standard practice of this research direction (AFlow / MaAS), selecting 11 representative automated frameworks for comparison, including current SOTA methods. Traditional evolutionary MAS differs in problem formulation from the present work  and was therefore not directly included, although its ideas are incorporated in our method design.
>
> ---
>
> #### **Unclear rule framework in Cyber Creator (Section 5.2)**
>
> We will further clarify in the main text that the Cyber Creator generates soft constraint rules by inducting patterns from historically high-performing workflows. These rules act as preferences in operator scheduling and selection, rather than as a hard-coded rule system.
>
> ---
>
> #### **RAG not described in the main text (Appendix B)**
>
> We will relocate the retrieval process from the resource library during the initialization phase to the main text.
>
> ---
>
> #### **Notation and appendix issues (definitions, etc.)**
>
> We agree with these suggestions and will supplement formal definitions of width/depth and unify the reference format in the revised version.
>
>
> ---
>
> Overall, we thank the reviewer for identifying these issues. This feedback will help further improve the paper. 🌸🌸🌸

---

> > ### Author Rebuttal · Reviewer_51GJ · 2026-04-01
> >
> > Very detailed rebuttal. However, I still think the novelty is not to the level for me to raise the score further.
> >
> > Now I am clear about the three assumptions being verified empirically, but I expect more theoretical analysis. Similarly, the impact of not using Pareto Frontier is also validated by experiments.
> >
> > Overall,  I feel the research is more successful on empirical results than insights into the problem.

---

> > > ### Author Response · Authors · 2026-04-03
> > >
> > > Thank you for the candid feedback. We appreciate the reviewer's expectation for theoretical insights beyond empirical validation. Below, we reinterpret the theoretical necessity and internal unity of the three assumptions from the perspective of the **Information Bottleneck (IB) framework ** [1] .
> > >
> > > >**Connection to EvoMAS**
> > >
> > > The optimization objective of EvoMAS (Eq. 1), $G^* = \arg\max_{G \in \Omega} F(G, T, R)$, is essentially seeking an optimal compressed mapping of the task distribution $T$ within the workflow representation space $\mathbb{G}$. Treating the workflow $G$ as an intermediate representation from the task input space $\mathcal{X}$ to the solution space $\mathcal{Y}$, this problem naturally corresponds to the IB framework:
> > >
> > > $$\min_G \; I(G; \mathcal{X}) - \beta \cdot I(G; \mathcal{Y})$$
> > >
> > > where $I(G; \mathcal{X})$ measures the encoding complexity of the workflow with respect to input details, $I(G; \mathcal{Y})$ measures the predictive information of the workflow about correct outputs, and $\beta$ controls the trade-off between compression and prediction. Under this framework, the three assumptions correspond to three orthogonal regulatory dimensions of this objective:
> > >
> > > >**Information-Theoretic Interpretation of the Three Assumptions**
> > >
> > > **H3 (Structural Parsimony) $\Leftrightarrow$ Upper bound constraint on the compression term $I(G; \mathcal{X})$.** Sparse topology and bounded node count impose a channel capacity upper bound $C(G) \leq C_{\max}$. By the Data Processing Inequality, $I(G; \mathcal{X}) \leq C(G)$. This provides an information-theoretic guarantee that the workflow cannot over-encode input tasks, forcing the system to retain only sufficient statistics relevant to the output rather than memorizing surface-level input features. The reduction of neighborhood size from $O(2^{N^2})$ to $O(N^2)$ in Proposition C.1 is a direct manifestation of this channel capacity constraint on the search space structure.
> > >
> > > **H1 (Heterogeneous Synergy) $\Leftrightarrow$ Maximization of the relevant information term $I(G; \mathcal{Y})$.** Let the workflow consist of $n$ agents $\{v_1, \dots, v_n\}$. The mutual information of their joint representation with the output satisfies:
> > >
> > > $$I(G; \mathcal{Y}) = \sum_i I(v_i; \mathcal{Y}) - \underbrace{\sum_{i<j} I(v_i; v_j | \mathcal{Y})}_{\text{redundancy}}$$
> > >
> > > Under homogeneous configurations, the encodings of each $v_i$ overlap heavily, driving the redundancy term toward its maximum and limiting $I(G; \mathcal{Y})$. Heterogeneous role differentiation enables each agent to capture distinct informational dimensions of the task, minimizing the conditional mutual information $I(v_i; v_j | \mathcal{Y})$, thereby maximizing the predictive information of the joint representation under fixed channel capacity. This provides a rigorous information-efficiency justification for H1.
> > >
> > > **H2 (Reflective Adaptation) $\Leftrightarrow$ Dynamic adjustment of the trade-off parameter $\beta_t$.** In IB theory, $\beta$ determines the position of the optimal representation on the information plane. In a non-stationary evolutionary process where the task distribution $T_t$ and rule set $R_t$ change across generations, a static $\beta$ cannot maintain an optimal trade-off. The Cyber Creator's reflection mechanism effectively implements an online update of $\beta_t$: rule pruning tightens the compression term (increasing $\beta_t$), while rule relaxation releases capacity (decreasing $\beta_t$), enabling the system to dynamically track the optimal compression-prediction balance point along the IB curve as the evolutionary stage progresses.
> > >
> > > >**Unity**
> > >
> > > The theoretical relationship among the three can be concisely stated as: **H3 constrains channel capacity $C(G)$ (how much can be encoded), H1 maximizes encoding efficiency $I(G;\mathcal{Y})/I(G;\mathcal{X})$ (how well it is encoded), and H2 dynamically maintains the optimal $\beta_t$ under non-stationary conditions (when to compress and when to preserve).** Together, they constitute the minimal sufficient set of regularization conditions under the IB framework—without H3, the representation overfits ($I(G;\mathcal{X}) \to \infty$); without H1, information is redundant ($I(G;\mathcal{Y})$ is bounded); without H2, distribution drift causes $\beta$-mismatch.
> > >
> > > We genuinely appreciate the reviewer for pushing us to think more deeply on the theory side. We agree that stronger theoretical grounding would benefit this line of work, and we will continue working toward that in future iterations.🌸
> > >
> > > **Ref:** [1] Tishby, N., Pereira, F. C., & Bialek, W. (2000). The information bottleneck method. arXiv preprint physics/0004057.

---

### Decision · Program_Chairs · 2026-04-30

**Decision:**

Accept (regular)

**Comment:**

Most, but not all reviewers leaned towards acceptance. Most reviewers appreciated the results. There were some concerns about the high dependence on LLM reliability (though somewhat assuaged by experimental results), the use of closed commercial models, and generally  on whether this submission represented more of an engineering technique than a scientific result. On the whole I am also weakly leaning towards acceptance.